

# Pan-Arctic measurements of wintertime water vapour column using a satellite-borne microwave radiometer

Christopher Perro[1], Thomas J. Duck[1], Glen Lesins[1], Kimberly Strong[2], Penny M. Rowe[3,4],
James R. Drummond[1], and Robert J. Sica[5]

[1]Dalhousie University, Halifax, Nova Scotia, Canada
[2]University of Toronto, Toronto, Ontario, Canada
[3]NorthWest Research Associates, Redmond, WA, USA
[4]Department of Physics, Universidad de Santiago de Chile, Santiago, Chile
[5]Western University, London, Ontario, Canada

**Correspondence:** Christopher Perro (christopher.perro@dal.ca)

.

**Abstract.** A methodology for retrieving high-latitude winter water vapour columns from passive microwave satellite measurements from Perro et al. (2016) is extended to use measured surface reflectance ratios under more realistic surface reflection assumptions. Pan-Arctic wintertime water vapour is retrieved from Advanced Technology Microwave Sounder (ATMS) measurements made from January 2012 through March 2015 (December to March). The water vapour retrievals are validated using two ground based instruments: the G-band Vapor Radiometer (GVR) at Barrow, Alaska, and the Extended-Range Atmospheric Emitted Radiance Interferometer (E-AERI) at Eureka, Nunavut. E-AERI was chosen as an additional point of validation compared to Perro et al. (2016) due to the different technology and frequencies employed to determine water vapour column compared to the ATMS and GVR. For water vapour columns less than $6\,\mathrm{kg\,m^{-2}}$, the biases are $+2.6\%$ and $+0.01\%$ relative to the GVR and E-AERI, respectively. A comparison with radiosonde humidity measurements shows they are dry relative to the ATMS measurements in North America and Western Europe, and moist in Asia and Eastern Europe, with an apparent dependence on radiosonde manufacturer. Reanalyses (ERA-5, ERA-Interim, ASR V2, JRA-55 and NCEP) are systematically drier than the ATMS measurements for water vapour columns less than $6\,\mathrm{kg\,m^{-2}}$, with relative biases ranging from $-10\%$ to $-23\%$. These differences could have implications for the understanding of the Arctic water budget and climate.

## 1 Introduction

Water vapour is an important greenhouse gas that contributes to global temperature increases through a strong feedback effect (Curry et al., 1995). In the Arctic, closure of the $20\,\mu$m infrared transmission window (Stamnes et al., 1999) is expected to enhance the feedback through mid-century (Chen et al., 2011). Long-term precision measurements of water vapour are therefore important for monitoring and understanding Arctic climate (Cox et al., 2015).

Surface-based water vapour measurements in the Arctic are few and spatially sparse. Although satellite measurements can fill the gaps, observations in the infrared are limited by cloud cover, which occurs with a frequency of nearly $50\%$ during the



winter season (Boccolari and Parmiggiani , 2018). Microwave measurements, on the other hand, are only weakly affected by clouds. Polar-orbiting passive microwave radiometers such as the Advanced Microwave Sounding Unit B (AMSU-B), Special Sensor Microwave Imager/Sounder (SSMIS), Microwave Humidity Sounder (MHS), Advanced Technology Microwave Sounder (ATMS), MicroWave Humidity Sounder (MWHS), and the Special Sensor Microwave/Temperature 2 (SSM/T-2) pro-

vide over 25 years of continuous measurements that can be used for water vapour monitoring. Measurements in the vicinity of the strong 183 GHz water vapour absorption line are particularly useful in the relatively dry conditions found in the Arctic during winter.

    Here we present pan-Arctic water vapour column obtained from wintertime satellite microwave measurements made with the ATMS between January 2012 and March 2015 (December to March inclusive). Water vapour columns were determined using

an updated version of the retrieval presented by Perro et al. (2016). The retrieval builds on the work of Miao et al. (2001) and Melsheimer and Heygster (2008) by employing auxiliary information for atmospheric conditions and numerical optimization. It was originally validated using measurements at Barrow, Alaska (71.3°N, 156.8°W) with prescribed surface emissivities and an assumption of specular surface reflection. Surface types vary across the Arctic in both space and time, and this must be taken into account in any pan-Arctic treatment. For this purpose, we improve the retrieval by using the surface reflection

properties and emissivity retrieval given by Perro et al. (2018) to determine the spatial and temporal surface properties, making the retrieval more suitable for high-latitude measurements.

    ATMS measurements from the Suomi-NPP satellite are used exclusively in this study. ATMS is the sucessor to MHS, the instrument considered in the study by Perro et al. (2016). All five 183 GHz dual-band channels, the 165.5 GHz channel and the 88.2 GHz channel of ATMS are used. See Table 1 for a summary of the frequencies and polarizations of interest.

In this paper, the ATMS water vapour columns are validated against surface-based measurements from two different instruments: the G-band Vapor Radiometer (GVR) at Barrow, Alaska (similar to the work of Perro et al. (2016)) and the Extended-Range Atmospheric Emitted Radiance Interferometer (E-AERI) at Eureka, Nunavut (80.0°N, 85.9°W). The ATMS-derived vater vapour columns are subsequently compared to those from the Arctic radiosonde network and multiple reanalyses, including the new ERA5 product from the European Centre for Medium-Range Weather Forecasts (ECMWF).

The ERA5 reanalysis water vapour product is found to be systematically drier than the ATMS measurements by 11% on average. Other reanalyses (ERA-Interim, JRA-55, ASR V2, and NCEP) are similarly dry. The reasons for this bias are not presently understood but are important to resolve. If reanalyses should prove too dry, then this would have implications for our understanding of the Arctic water vapour budget, radiative transfer and possibly climate.

    The structure of this paper is as follows. Section 2 describes the updates made to the microwave water vapour retrieval

of Perro et al. (2016) for the purposes of high-latitude measurements (some details are relegated to Appendix A). In Sec. 3 the retrieval is validated against the GVR and E-AERI. Section 4 provides comparisons with measurements from the Arctic radiosonde network and Sec. 5 shows comparisons with ERA5, ERA-Interim, JRA-55, ASR V2 and NCEP reanalyses. Conclusions are presented in Sec. 6.



## 2 Water Vapour Column Retrieval

### 2.1 Formulation

The water vapour retrievals of Miao et al. (2001), Melsheimer and Heygster (2008) and Perro et al. (2016) all begin with the radiative transfer parameterization of Guissard and Sobieski (1994). The brightness temperature $T_i$ measured at frequency $\nu_i$ by channel $i$ of a satellite-borne microwave radiometer is

$$T_i = m_\mathrm{p}(\nu_i)T_\mathrm{s} - (T_\mathrm{o} - T_\mathrm{c})(1 - \varepsilon_i)\,t_i^2 \tag{1}$$

where $T_\mathrm{s}$ is the skin temperature, $T_\mathrm{o}$ is the surface air temperature, $T_\mathrm{c}$ is the cosmic background temperature, $\varepsilon_i$ is the surface emissivity, and $t_i$ is the slant transmission. The factor $m_\mathrm{p}$ incorporates the vertical structure of the atmosphere, and is given by

$$
\begin{aligned}
m_p(\nu_i) = {} & 1 + (1 - \varepsilon_i t_i)\frac{T_\mathrm{o} - T_\mathrm{s}}{T_\mathrm{s}} \\
& - \frac{1}{T_\mathrm{s}}\left(\int_0^\infty -\left(1 - t_i(z,\infty)\right)\frac{\mathrm{d}T}{\mathrm{d}z}\mathrm{d}z + (1 - \varepsilon_i)t_i^2\int_0^\infty \left(1 - t_i^{-1}(z,\infty)\right)\frac{\mathrm{d}T}{\mathrm{d}z}\mathrm{d}z\right).
\end{aligned}
\tag{2}
$$

where $t_i$ in Eqs. (1) and (2) is given by

$$t_i(z_1, z_2) = e^{-\tau_i(z_1, z_2)\sec\theta} \tag{3}$$

where $\tau(z_1, z_2)$ is the zenith optical depth between altitudes $z_1$ and $z_2$. In Eqs. (1) and (2), the transmission $t_i$ given without arguments implies $t_i(0, \infty)$.

In the above formulation, the surface reflection is assumed to be specular so that $\theta$ is the zenith angle for both the downwelling and upwelling radiation paths. In this work we consider both specular and Lambertian reflection at the surface, consistent with the surface reflectivity results of Perro et al. (2018). For Lambertian reflection, the effective incident angle (Matzler, 2005) is used as the zenith angle for the downwelling path. Following the derivation by Guissard and Sobieski (1994), we rewrite Eq. (1) as

$$T_i = T_\mathrm{s}m_\mathrm{p} - (T_\mathrm{o} - T_\mathrm{c})(1 - \varepsilon_i)t_{i,\mathrm{D}}t_{i,\mathrm{U}} \tag{4}$$

and the factor $m_\mathrm{p}$ as

$$
\begin{aligned}
m_\mathrm{p}(\nu_i) = {} & 1 + (1 - \varepsilon_i t_{i,\mathrm{U}})\frac{T_\mathrm{o} - T_\mathrm{s}}{T_\mathrm{s}} \\
& - \frac{1}{T_\mathrm{s}}\Bigg(\int_0^\infty -\left(1 - t_{i,\mathrm{U}}(z,\infty)\right)\frac{\mathrm{d}T}{\mathrm{d}z}\mathrm{d}z \\
& \qquad + (1 - \varepsilon_i)t_{i,\mathrm{D}}t_{i,\mathrm{U}}\int_0^\infty \left(1 - t_{i,\mathrm{D}}^{-1}(z,\infty)\right)\frac{\mathrm{d}T}{\mathrm{d}z}\mathrm{d}z\Bigg)
\end{aligned}
\tag{5}
$$

where $t_{i,\mathrm{U}}$ and $t_{i,\mathrm{D}}$ are the transmittances for the slant upwelling and downwelling radiation paths, respectively.

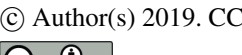



Following the derivation by Miao et al. (2001), the brightness temperatures at three different frequencies $T_1$, $T_2$, and $T_3$ may be combined to write

$$\frac{\Delta T_{12} - b_{12}}{\Delta T_{23} - b_{23}} = (r_1/r_2) \left( \frac{t_{1,\mathrm{D}}t_{1,\mathrm{U}} - (r_2/r_1)t_{2,\mathrm{D}}t_{2,\mathrm{U}}}{t_{2,\mathrm{D}}t_{2,\mathrm{U}} - (r_3/r_2)t_{3,\mathrm{D}}t_{3,\mathrm{U}}} \right) \tag{6}$$

where $\Delta T_{12} = T_1 - T_2$, $\Delta T_{23} = T_2 - T_3$. Surface reflectances for the three different frequencies are represented by $r_1$, $r_2$, and $r_3$ and the bias coefficients $b_{12}$ and $b_{23}$ are given by

$$b_{ij} = \int_0^\infty (t_{j,\mathrm{U}}(z,\infty) - t_{i,\mathrm{U}}(z,\infty)) \frac{\mathrm{d}T(z)}{\mathrm{d}z} \mathrm{d}z + (T_\mathrm{o} - T_\mathrm{s})(\varepsilon_j t_{j,\mathrm{U}} - \varepsilon_i t_{i,\mathrm{U}}) +$$

$$r_j t_{j,\mathrm{D}} t_{j,\mathrm{U}} \int_0^\infty \left(1 - t_{j,\mathrm{D}}^{-1}(z,\infty)\right) \frac{\mathrm{d}T(z)}{\mathrm{d}z} \mathrm{d}z - r_i t_{i,\mathrm{D}} t_{i,\mathrm{U}} \int_0^\infty \left(1 - t_{i,\mathrm{D}}^{-1}(z,\infty)\right) \frac{\mathrm{d}T(z)}{\mathrm{d}z} \mathrm{d}z. \tag{7}$$

These equations may be compared with Eqs. (2) and (3) in Perro et al. (2016) for the purely specular case. As in Perro et al. (2016), Eq. (7) was simplified by neglecting the term proportional to the difference in surface and skin temperatures, and supposing $r = r_i = r_j$ is a constant so that

$$b_{ij} \approx \int_0^\infty (t_{j,\mathrm{U}}(z,\infty) - t_{i,\mathrm{U}}(z,\infty)) \frac{\mathrm{d}T(z)}{\mathrm{d}z} \mathrm{d}z + r \left[ t_{j,\mathrm{D}} t_{j,\mathrm{U}} \int_0^\infty \left(1 - t_{j,\mathrm{D}}^{-1}(z,\infty)\right) \frac{\mathrm{d}T(z)}{\mathrm{d}z} \mathrm{d}z \right.$$

$$\left. - t_{i,\mathrm{D}} t_{i,\mathrm{U}} \int_0^\infty \left(1 - t_{i,\mathrm{D}}^{-1}(z,\infty)\right) \frac{\mathrm{d}T(z)}{\mathrm{d}z} \mathrm{d}z \right]. \tag{8}$$

Equations (6) and (8) are solved for the water vapour column given microwave satellite brightness temperatures in the same manner as Perro et al. (2016). A numerical nonlinear optimizer is used with the aid of auxiliary water vapour shape and temperature profiles. A radiative transfer model is required to determine the optical depth profiles, which depend on the vertical distribution of water vapour and other constituents. Similar to Perro et al. (2016, 2018), we use the RTTOV 1-D radiative transfer model (Matricardi and Saunders, 1999).

Retrievals using Eqs. (6) and (8) reveal a dependency of the measured water vapour column on satellite local zenith angle, and discontinuities when the satellite channel selection is changed for measurements in different water vapour column regimes (a topic further discussed in Sec. 2.3). These problems likely stem from systematic errors in the auxiliary profiles, approximations simplifying Eq. (8), and systematic measurement errors. To correct them, we introduce tuneable parameters $\delta b_{23}$, $\delta \frac{r_1}{r_2}$, and $\delta \frac{r_2}{r_3}$ in Eq. (6) to give

$$\frac{\Delta T_{12} - b_{12}}{\Delta T_{23} - (b_{23} + \delta b_{23})} = \left( \frac{r_1}{r_2} + \delta \frac{r_1}{r_2} \right) \left( \frac{t_{1,\mathrm{D}}t_{1,\mathrm{U}} - \left( \frac{r_1}{r_2} + \delta \frac{r_1}{r_2} \right)^{-1} t_{2,\mathrm{D}}t_{2,\mathrm{U}}}{t_{2,\mathrm{D}}t_{2,\mathrm{U}} - \left( \frac{r_2}{r_3} + \delta \frac{r_2}{r_3} \right)^{-1} t_{3,\mathrm{D}}t_{3,\mathrm{U}}} \right). \tag{9}$$





Each parameter is assumed to correct a systematic error in the given variable or ratio of variables. A final adjustment, $\delta W$, is applied to the water vapour column in the mid and extended regimes (see Sec. 2.3). The methods used to calibrate these parameters are described in Appendix A. The calibration is purely internal and so does not depend on any outside data source.

For this study, ERA-Interim reanalyses provided the auxiliary profiles, and RTTOV 12.1 was used in the retrieval. These

choices led to specific calibrations for the tuneable parameters. Although ERA5 has now been released, the retrieval code was frozen earlier against external updates in order to move past the calibration stage and onto data processing and analysis. Recent updates to either the auxiliary profiles or the RTTOV version are not expected to impact the conclusions of this study.

## 2.2   Surface Reflection Mixtures

Perro et al. (2018) showed for microwave frequencies that land, first-year ice (FYI) and multi-year ice (MYI) may be treated

as Lambertian reflectors, while the ocean is better represented as a specular reflector. Testing has shown, however, that water vapour retrieval errors can be minimized by treating the ocean as a mixed specular-Lambertian reflector (Perro , 2017). Reflections from ocean waves are assumed to be the source of the Lambertian component.

A mixture of specular and Lambertian surface reflection may be represented as a linear average of specular and Lambertian brightness temperatures

$$T_{\mathrm{mix}} = S T_{\mathrm{specular}} + (1 - S) T_{\mathrm{Lambertian}} \tag{10}$$

where $S$ is the fraction of specular reflection, and $T_{\mathrm{specular}}$, $T_{\mathrm{Lambertian}}$ and $T_{\mathrm{mix}}$ are the brightness temperatures for specular, Lambertian, and mixed reflection, respectively (Matzler, 2005). The effective zenith angle of the downwelling path $\theta_{\mathrm{D}}(S)$ can be expected to take on a value between the effective angle for Lambertian reflection $\theta_{\mathrm{D}}(S=0)$, and the satellite local zenith angle $\theta_{\mathrm{U}} = \theta_{\mathrm{D}}(S=1)$. Land, FYI, and MYI surfaces have $S = 0$ since they are considered Lambertian reflectors while ocean

has $S = 0.5$ due to it being treated as a mixed specular-Lambertian reflector.

The dependence of $\theta_{\mathrm{D}}(S)$ on $\theta_{\mathrm{U}}$ was determined using simulations. We used radiosonde measurements from Barrow, Alaska as input data and generated brightness temperatures with RTTOV by assuming a surface emissivity of 0.8 for all frequencies, cloudless skies, unpolarized radiation, and equal surface air and skin temperatures. Simulations were produced for the ATMS instrument for a range of satellite zenith angles. The effective angle $\theta_{\mathrm{D}}(S)$ was determined by minimizing the water vapour

column bias with respect to the model input for each value of $\theta_{\mathrm{U}}$.

Results for $S = 0.5$ are shown in Fig. 1. As expected, the values for $\theta_{\mathrm{D}}(S=0.5)$ fall between those for $\theta_{\mathrm{D}}(S=0)$ and $\theta_{\mathrm{D}}(S=1)$ across the entire range of $\theta_{\mathrm{U}}$ values. A formula was fit to the results to obtain intermediate values.

## 2.3   Regimes

The retrieval is applied to ATMS measurements for different water vapour ranges, or water vapour *regimes*, in a manner similar

to that presented for MHS measurements by Perro et al. (2016) and for AMSU-B measurements by Melsheimer and Heygster (2008). Table 2 shows the channels used in each regime along with the slant water vapour column ranges. Separating regimes according to slant water vapour columns reduces errors in comparison with other techniques (Perro et al., 2016).





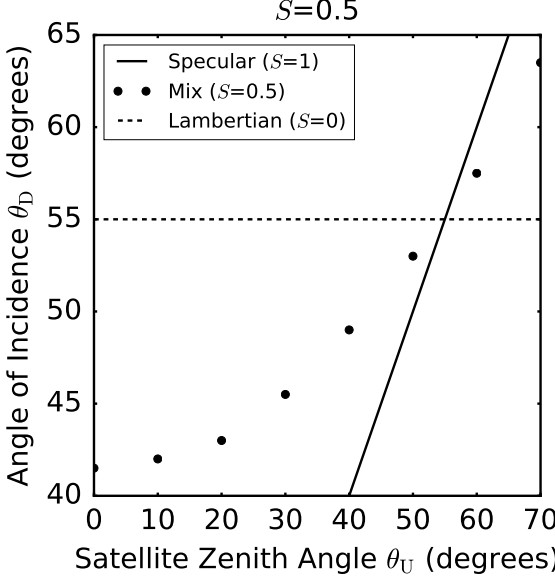

**Figure 1.** $\theta_D$ values determined from simulations ($S$=0.5) by minimizing water vapour column bias of our retrievals with model input at varying satellite local zenith angles.

Three regimes are used, and are labeled as *low*, *mid*, and *extended*. The slant water vapour column range for the *low* regime is 0 to $2.5\,\mathrm{kg\,m^{-2}}$, the *mid* regime is 1.5 to $10\,\mathrm{kg\,m^{-2}}$, and the *extended* is 9 to $15\,\mathrm{kg\,m^{-2}}$. The frequencies are chosen to be close to those used previously for MHS. The mid regime range was extended by $1\,\mathrm{kg\,m^{-2}}$ compared to the regimes from Perro et al. (2016) owing to improved performance of the updated retrieval with ATMS measurements. Weighted averages are used in the final data product where regimes overlap in order to smooth the transition (Perro et al., 2016).

ATMS has two additional channels with frequencies near $183\,\mathrm{GHz}$. Measurements from these two channels are not used due to the increased retrieval noise with small differences in frequency (which leads to similar brightness temperatures).

## 2.4 Surface Reflectance Ratios

Consistent with the approach of Miao et al. (2001) and Melsheimer and Heygster (2008), we assume the reflectances are equal for channels measuring near $183.31\,\mathrm{GHz}$. For both $88.2\,\mathrm{GHz}$ and $165.5\,\mathrm{GHz}$, the reflectances have different values. This corresponds to $r_1 = r_2 = r_3$ in the low regime, $r_1 \neq r_2 = r_3$ in the mid regime, and $r_1 \neq r_2 \neq r_3$ for the extended regime. Consequently, spatially-resolved values of $\frac{r_1}{r_2}$ are required for the mid and extended regimes, while values for $\frac{r_2}{r_3}$ are only required for the extended regime.

The surface reflectance retrieval used in this study is that of Perro et al. (2018). Land, FYI, MYI and ocean are identified using EUMETSAT Ocean and Sea Ice Satellite Application Facility ice type product (OSI-403-c; Breivik et al. (2012)). Land, FYI and MYI are treated as Lambertian reflectors, whereas water surfaces (oceans) are taken to be mixed ($S = 0.5$) Lambertian-specular reflectors.





Zenith reflectance ratio maps are produced weekly using the median reflectance ratio measured in the surrounding two weeks of overpasses. To maximize data usage, measurements at oblique angles are adjusted for zenith viewing by using the emissivity variation with viewing angle from Perro et al. (2018) to form transfer functions. The same transfer functions are used to convert zenith reflectance ratios to the specific satellite zenith angle needed for a given water vapour retrieval.

Large water vapour columns over ocean can introduce significant errors in reflectance ratios, even during the relatively dry polar winter. For this reason, a reflectance ratio over water was taken from the winter season measurements of Perro et al. (2018). The value only varies with satellite zenith angle and is constant spatially and temporally.

In most cases, surface reflectivity measurements are limited to having slant water vapour columns less than $3\,\mathrm{kg\,m^{-2}}$ at each location to reduce the impact of auxiliary information on the retrievals. If the number of measurements at a location is
lower than a threshold (10), then the limit is extended to $5\,\mathrm{kg\,m^{-2}}$. Climatological values from Perro et al. (2018) are used at locations where the threshold cannot be met. Climatological values are used more frequently in regions of relatively high water vapour column, such as over Europe, and less so over drier areas, such as the Canadian Arctic.

## 3   Validation against Surface-Based Measurements

### 3.1   Instruments

The G-band Vapor Radiometer (GVR) operated in Barrow, Alaska, is used as the first validation source. Comparisons of the ATMS-derived water vapour column to values retrieved from the GVR follow along the lines of the work from Perro et al. (2016), although here we assume Lambertian reflection at the surface and measured – *not prescribed* – surface reflectivities for the satellite retrieval.

The GVR remotely senses water vapour from the surface using dual-band channels at 183±14,7,3,1 GHz; i.e., in the same
spectral range as ATMS. Water vapour columns are determined using a neural-network algorithm trained using precision radiosonde measurements (Cadeddu et al., 2009). Three-minute averages of the $4\,\mathrm{min^{-1}}$ GVR measurements (Pazmany, 2007) were used to reduce noise.

The Extended-Range Atmospheric Emitted Radiance Interferometer (E-AERI) is a surface-based instrument located in Eureka, Nunavut, which is used as a second validation source. The E-AERI is a Fourier Transform Infrared Spectrometer mea-
suring in the 3–25 $\mu$m range (Mariani et al., 2012); i.e., in a completely different spectral region from ATMS and the GVR. Seven minute temporal resolution is used in the comparison.

The E-AERI retrieval is described in Rowe et al. (2008) and in Weaver et al. (2017). The retrieval makes use of twice-daily radiosoundings as well as surface-based measurements of trace gases (e.g. $CO_2$) to determine the atmospheric state and the shape of the humidity profile. Cases containing radiatively-significant clouds are screened out (Weaver et al., 2017) and water
vapour column is retrieved.



## 3.2 Results

Figure 2a shows the distribution of ATMS-derived water vapour columns against GVR measurements from January 2012 to March 2015 for December through March inclusive. Measurements within 50 km of Barrow, Alaska were used, which resulted in 3237 individual comparisons. No temporal coincidence constaint between the GVR and ATMS overpasses was required

since the GVR was running continuously during the times of the ATMS overpasses.

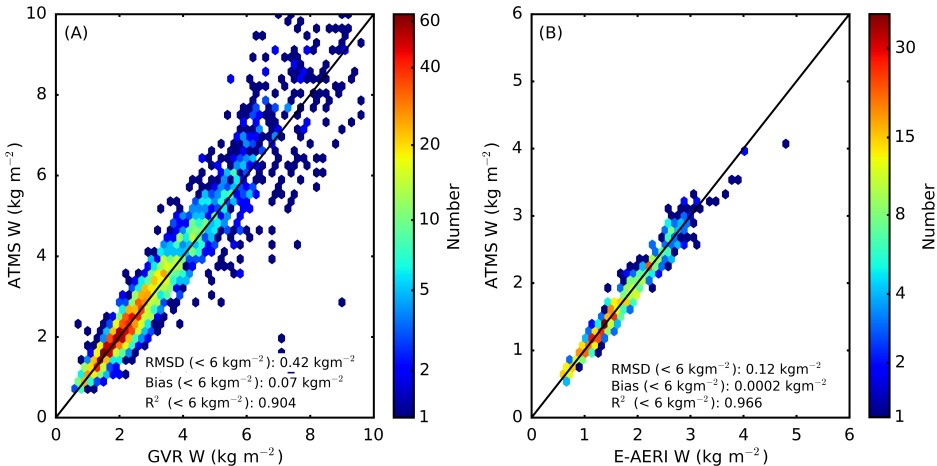

**Figure 2.** ATMS-derived water vapour column ($W$) versus (A) GVR measurements at Barrow, Alaska and (B) E-AERI measurements at Eureka, Nunavut between January 2012 and March 2013 (December to March inclusive). Number of cases are given in the logarithmic colour scale. Positive biases correspond to relative moistness of the ATMS-derived measurements. Biases are indicated on each panel; positive biases correspond to relative moistness of the ATMS-derived measurements. The black line is the 1:1 line.

The RMS deviation and bias relative to the GVR were calculated for columns less than $6\,\mathrm{kg\,m^{-2}}$, consistent with the study of Perro et al. (2016). The RMS deviation is significantly larger for water vapour columns greater than $6\,\mathrm{kg\,m^{-2}}$ due to noise in the extended regime. For the comparison with the GVR at Barrow, the RMS deviation is $3.94$ times larger for water vapour columns between $6\,\mathrm{kg\,m^{-2}}$ and $10\,\mathrm{kg\,m^{-2}}$ compared to water vapour columns less than $6\,\mathrm{kg\,m^{-2}}$.

For water vapour columns less than $6\,\mathrm{kg\,m^{-2}}$, the RMS deviation and bias relative to the GVR are $0.42$ and $+0.07\,\mathrm{kg\,m^{-2}}$, respectively (the ATMS-derived columns are slightly moister). These results are nearly the same as those provided by Perro et al. (2016), despite the different instrument and smaller number of measurements considered in this case. The relative RMS deviation and bias are $15\%$ and $+2.6\%$, respectively. The relative differences were calculated by dividing the RMS deviation and bias by the mean of the water vapour column (less than $6\,\mathrm{kg\,m^{-2}}$) of the comparator data product (in this case, GVR

measurements). This method for calculating relative differences is used throughout this work.

There are several aspects of the retrieval that are improved which are not represented in the RMS deviation or bias. The effect of satellite zenith angle is minimized, as are differences between regimes and the decreasing trend between nadir and oblique angle measurements (see Appendix A). Also, the reflectance ratio maps allow the retrieval to be used at various locations across



the Arctic. These differences make the retrieval more consistent across a range of satellite zenith angles, water vapour columns, and locations.

Figure 2B shows the distribution of ATMS-derived water vapour columns against E-AERI measurements from January 2012 to March 2015 during the winter season (December to March inclusive). A total of 789 measurements is used in the comparison,

which considered ATMS measurements within a $50\,\mathrm{km}$ radius with a maximum time difference of $1\,\mathrm{h}$. The relatively small number of measurements in the comparison is due to the removal of measurements containing radiatively significant clouds Weaver et al. (2017). For water vapour columns less than $6\,\mathrm{kg\,m^{-2}}$, the RMS deviation and bias are $0.12$ and $+0.0002\,\mathrm{kg\,m^{-2}}$, respectively (the ATMS-derived columns are slightly moister). These correspond to relative differences of $7.2\%$ and $+0.01\%$, respectively. While measurements from ATMS and the E-AERI compare extremely well, it should be noted that the range of

water vapour values encountered is significantly smaller than for the GVR.

The terrain at Eureka is different from that at Barrow. At Eureka, there is a large amount of sloping terrain while at Barrow, the terrain is fairly flat and near sea level. The sloping terrain can potentially influence the water vapour columns obtained using our retrieval. It is difficult to isolate and therefore evaluate the effect it has on the ATMS-derived water vapour columns. However, the level of agreement of the ATMS water vapour retrieval with both the GVR and E-AERI suggest that effects

associated with terrain are not a significant issue.

The agreement between the ATMS and surface-based measurements provides confidence that the ATMS measurements are valid at other locations in the Arctic.

## 4   Radiosondes Comparison

Figure 3A and 3B provide a comparison between ATMS-derived water vapour columns and those from the global operational

radiosonde network. Both the relative RMS deviation and bias are shown. ATMS overpasses within 50 km and 3 hours from a radiosonde launch were used in the comparison, which spans between January 1, 2012 and March 31, 2015 for the winter months (December to March). Radiosonde measurements were obtained from the Integrated Global Radiosonde Archive (IGRA) (Durre et al., 2006). IGRA applies a quality control procedure to remove erroneous values in the radiosonde measurements. Radiosondes water vapour column was determined using profiles ranging from the surface to an altitude of 12

km.

The relative RMS deviation ranges from about 11 to 30% across the Arctic. The small relative RMS deviations in Northern Canada are associated with generally small water vapour columns. The outlier in Iceland is likely associated with a large number of the satellite overpasses being over ocean where there are relatively larger water vapour columns in comparison with the rest of the Arctic.

Measurements of relative bias, on the other hand, show systematic trends depending upon region. The satellite measurements are systematically moister in the North American and Western European sector, and drier in the Eastern European/Asian sector. Figure 3C shows the radiosonde instrument primarily used at each station. The systematic moistness of the radiosonde measurements in the Eastern European/Asian sector corresponds to Russian-manufactured radiosondes. A moist bias for Russian-



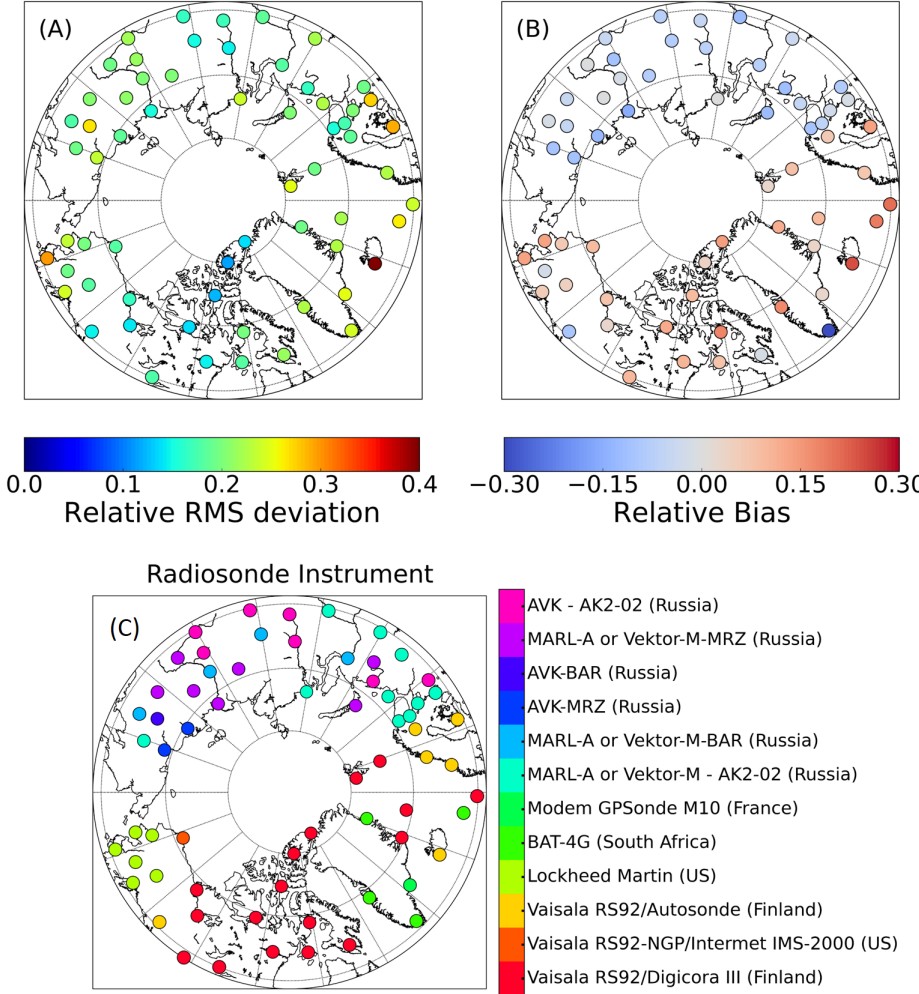

**Figure 3.** (A) Relative RMS deviation and (B) biases between ATMS-derived and radiosonde water vapour columns at 62 Arctic launch stations for water vapour column less than $6 \, \mathrm{kg \, m^{-2}}$ between January 2012 and March 2013 (December to March inclusive). Positive biases correspond to relative moistness of the ATMS-derived measurements compared to the radiosondes. Also, a (C) Pan-Arctic map of radiosonde instruments that are predominantly used at each station. The relative differences were calculated by dividing the RMS deviation and bias by the mean of the water vapour column of the comparator data product.

manufactured scientists is in agreement with prior work by Morardi et al. (2013), who used satellite brightness temperatures from AMSU-B and MHS, and radiative transfer simulations globally to find a moist bias for Russian-manufactured radiosondes, and a dry bias elsewhere.





## 5 Reanalyses Comparison

Figure 4 shows the number distribution of ATMS-derived water vapour columns against ERA5 values. Measurements were chosen from 711 locations in a regular grid with $2.5°$ latitude and $5°$ longitude north of $60°$, except for latitudes greater than 80°N where the longitudinal resolution was reduced with increasing latitude. ATMS overpasses within 50 km of each grid point and with a maximum time difference of 3 h were used between January 1, 2012 and March 31, 2015 for December through March inclusive. The comparison includes a total of 2,194,436 measurements.

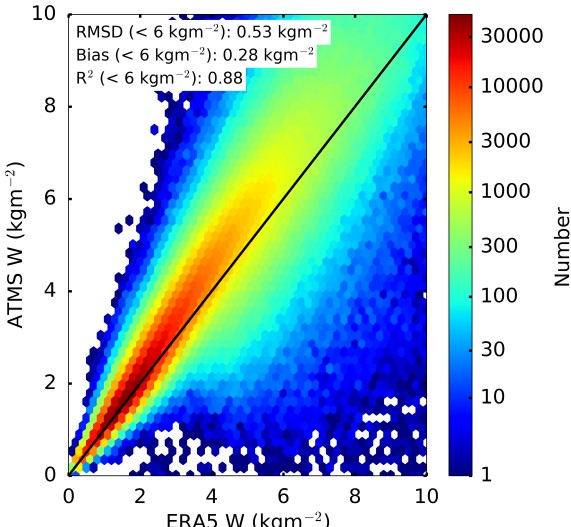

**Figure 4.** Number distributions for ATMS-derived water vapour column ($W$) against ERA5 values at 711 locations between January 2012 and March 2015 (December to March inclusive). Relative RMS deviations and biases are given for ERA5 water vapour columns less than $6\,\mathrm{kg\,m^{-2}}$. Positive biases correspond to relative moistness of the ATMS-derived columns relative to the ERA5 water vapour columns.

As was the case for the comparisons to E-AERI and GVR, a high level of correlation is observed. For water vapour columns less than $6\,\mathrm{kg\,m^{-2}}$, the RMS deviation and bias are 0.53 and $+0.28\,\mathrm{kg\,m^{-2}}$, respectively (the ATMS-derived columns are slightly moister). These correspond to relative differences of $22\%$ and $+11\%$, respectively (the ERA values are drier). However, the bias is considerably larger than what was found for the comparisons to E-AERI and GVR.

RMS deviations and biases for other reanalyses are given in Table 3, which also includes results from comparisons with the GVR, E-AERI, and radiosondes that were previously discussed. For the renalyses, the RMS deviations are smallest for ERA5 and the bias was smallest for ERA-Interim.

Figure 5 shows the spatial distribution of the relative RMS deviation and bias with respect to ERA5. The relative RMS deviation is fairly constant over most of Russia, Greenland, Canada, and sea ice. There are larger values in Alaska, Europe, along the coast of Greenland, and near the boundary of sea ice and open ocean.

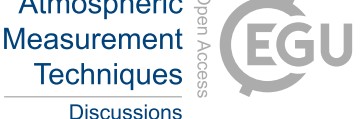



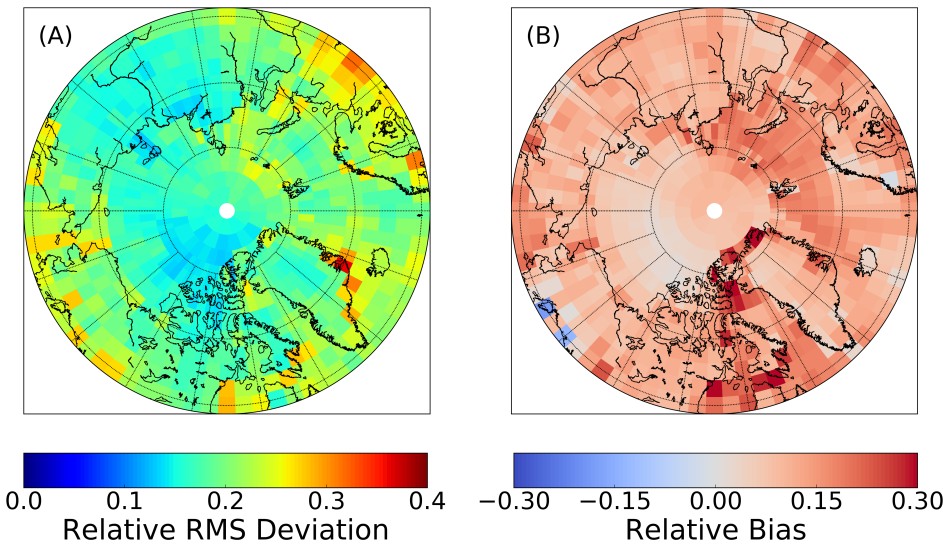

**Figure 5.** Relative RMS deviations and biases between ATMS-derived and ERA5 reanalysis water vapour columns at 711 locations with columns less than $6\,\mathrm{kg\,m^{-2}}$ between January 2012 and March 2013 (December to March inclusive). Positive biases correspond to relative moistness of the ATMS-derived measurements. The relative differences were calculated by dividing the RMS deviation and bias by the mean of the water vapour column of the comparator data product.

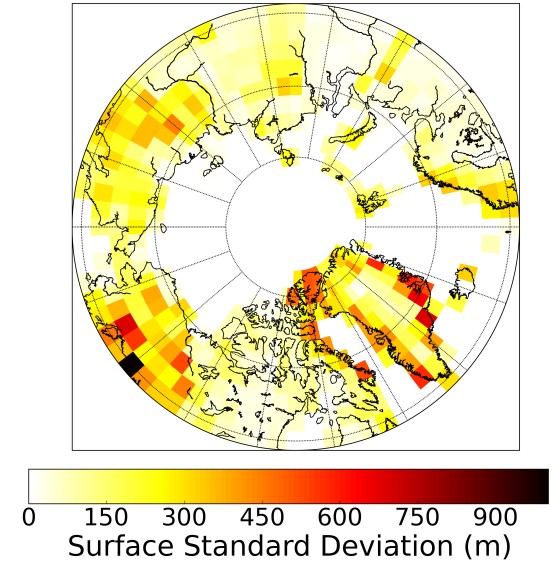

**Figure 6.** Standard Deviation of surface elevation for each of the 711 locations. Standard deviation is calculated for a footprint with a radius of 70 km.

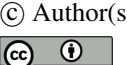



Figure 6 shows the surface standard deviation or surface elevation variability for each of the 711 locations chosen for the comparison with reanalyses. Comparing Fig. 5 with Fig. 6 shows that greater RMS deviations in Alaska, coastal Greenland, and Europe coincide with areas of significant surface elevation variability. Anomalous RMS deviations over the ocean and western Russia, however, are not associated with surface elevation variability. Also, there are regions with larger surface variability

that don't show relatively large RMS deviations, such as part of eastern Russia. The greater relative RMS deviations near the boundary of ocean and sea ice in the Atlantic Ocean are likely due to mixed surface types that are not addressed well enough in the retrieval.

The dry bias of ERA5 relative to the satellite measurements is found for all surface types and across most of the Arctic, except for a small number of isolated locations. Southern Greenland and Southern Alaska, for example, have a few locations

with relatively low biases which coincide with high surface elevation variability. In contrast, coastal northern Canada and Greenland have regions with relatively large biases which coincide with surface elevation variability. There are elevated biases near the boundary of sea-ice and ocean in the Atlantic ocean. This boundary is characterized by a gradual transition from ice to water that is not treated as a separate surface type in our analysis.

A number of locations are shown where the surface potentially influenced the comparison between ATMS-derived water

vapour columns and reanalyses. For regions with large surface elevation variability, surface scattering may not be character-ized as specular, Lambertian, or a linear mixture, which could influence ATMS-derived water vapour columns. Also, surface elevation variability may not be the most optimal quantity to compare to for characterizing anomolous RMS deviations and biases. Currently, we do not understand the reasons for the discrepancies that could not be described by surface elevation variability.

Previous studies have also found a dry bias in reanalysis data sets. ERA-40 reanalyses were found to be dry relative to AMSU-B-derived water vapour columns by Melsheimer and Heygster (2008) for February 2002. However, they assumed this was due to not accounting for reflectance ratios between 150 GHz and 183 GHz. This issue with the retrieval was fixed by Scarlat et al. (2018) who found that the ERA-Interim product was still drier than their AMSU-B-derived water vapour columns for all measurements north of $50°N$ between 2007 and 2009 during winter months (December and March).

## 25 6 Conclusions

In this work, the microwave satellite retrieval of water vapour given by Perro et al. (2016) was adapted for use over different high latitude surfaces. Careful attention was paid to surface scattering properties following their measurement by Perro et al. (2018). Land, FYI and MYI were treated as Lambertian reflectors, whereas water surfaces were treated as mixed Lambertian-specular reflectors. Reflectance ratio maps replaced the constant reflectance ratios used by Perro et al. (2016) for the mid and

extended regimes. Three tuneable parameters were included to reduce differences between water vapour columns between regimes and for different satellite zenith angles.

The ATMS-derived water vapour product was validated with measurements from the GVR in Barrow, Alaska, and the E-AERI in Eureka, Nunavut. Validation with the E-AERI instrument is novel to this work and provides increased confidence in



the ATMS results, particularly as the E-AERI operates in a completely different spectral region from ATMS and the GVR. Comparisons with the GVR and E-AERI showed good agreement with relative biases of $+2.6\%$ and $+0.01\%$ respectively for water vapour columns less than $6\,\mathrm{kg\,m^{-2}}$. RMS deviations were $15\%$ and $7.2\%$, respectively.

Comparisons of the ATMS-derived water vapour product were made against the Arctic radiosonde network and several reanalysis products including the new ERA5 reanalysis product from ECMWF. Compared to our measurements, radiosondes were relatively dry in North America and Western Europe, but relatively moist in Asia and Eastern Europe. This difference was attributed to regional differences in radiosonde manufacturers. Reanalyses were found to be dry compared to ATMS-derived water vapour column with ERA5 being $11\%$ drier on average.

The finding that reanalysis water vapour columns are too dry in the Arctic is consistent with findings from other microwave satellite-based studies. The validation against ground station measurements presented in this study improve confidence in the satellite-based retrievals, and provides further evidence for the dryness of reanalyses in the Arctic. Further validation from surface-based instruments is needed to ensure accuracy of the satellite retrievals over different surface types and under different conditions.

The discrepancy between reanalyses and ATMS measurements could have implications for our understanding of the Arctic atmospheric water budget and climate through the effect of water vapour on radiative transfer. The importance of this effect will need to be assessed by way of numerical calculations that are outside the scope of the current work.

The retrieval presented in this paper can also be applied to Antarctic measurements. A water vapour column product derived from the 25 years of continuous $183\,\mathrm{GHz}$ microwave measurements from ATMS, MHS, MWHS, SSMIS, AMSU-B, and SSM/T-2 would benefit polar research, and is left for future work.

*Data availability.* Retrieved water vapour column amounts are available on request from the corresponding author, Christopher Perro (christopher.perro@dal.ca)



## A  Empirical Corrections

Internal consistency checks for ATMS-derived water vapour column product show systematic disagreements. For example, uncorrected measurements at oblique satellite local zenith angles are drier than those from nadir. It is also found that regions of overlap between the different regimes disagree. These disagreements can be corrected using the tuneable parameters of Eq.

(9). The methods used to tune the parameters are described in this Appendix.

The errors are corrected in a three-step process: the bias coefficient adjustment parameter $\delta b_{23}$ is tuned first, followed by the reflection ratio adjustment parameters $\delta \frac{r_1}{r_2}$ and $\delta \frac{r_2}{r_3}$, and finally the water vapour column adjustment parameter $\delta W$. The calibration is purely internal in that it does not depend on any outside data sources.

The values for each adjustment parameter depend upon the satellite instrument used, and even to some degree on the version

of the radiative transfer model employed. The current procedure is manually intensive, a problem that will be remedied in future work. A summary of the values determined for use in this paper are given in Table 4.

### A.1  Bias Coefficient Adjustment Parameter

Figure 7 shows the mean total water vapour column differences between oblique ($40°$-$60°$) and nadir ($0°$-$20°$) satellite local zenith angles as a function of total column water for the low regime only. The differences were calculated at 711 locations, using

the regular grid described in Sec. 5. A maximum spatial difference of $50 \, \mathrm{km}$ from a location, and a maximum time difference of $3 \, \mathrm{h}$ between measurements, were allowed so as to minimize the impact of geophysical variability. The measurements are divided into four different surface types: land, first-year ice (FYI), multi-year ice (MYI), and ocean. Each series of curves represents different choices for the bias coefficient adjustment parameter, $\delta b_{23}$.

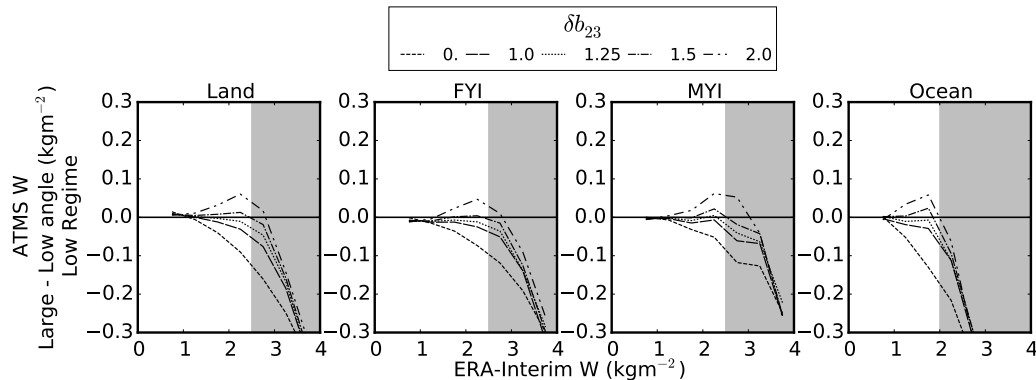

**Figure 7.** Mean differences in the low regime for ATMS-derived total water vapour columns between oblique ($40°$-$60°$) and nadir ($0°$-$20°$) satellite local zenith angles for different choices of $\delta b_{23}$. Gray shading indicates water vapour columns that are outside of the regime of interest.





The curves with $\delta b_{23} = 0$ in Fig. 7 are the mean differences between oblique and nadir zenith angles with no correction applied. For perfect agreement, the mean differences should be zero. What is observed with no correction applied, however, is that the differences become increasingly large for greater water vapour column values. This implies that measurements at oblique angles are increasingly dry compared to their nadir counterparts as water vapour column increases.

Simulations (Perro , 2017) indicate that errors in the bias coefficient $b_{23}$ can induce the above trends. The calculation of $b_{23}$ (Eq. (6)) depends on auxiliary water vapour shape and temperature profiles, which are obtained from reanalyses. Errors in reanalyses will therefore translate to errors in retrieval bias coefficients, which may be corrected using the bias coefficient adjustment parameter $\delta b_{23}$.

Different values for $\delta b_{23}$ give curves with different slopes in Fig. 7 . The choice that flattens the curve best in the region
of interest (white background) is selected. This choice is made (rather than, say, minimizing the difference) so as to avoid introducing systematic biases with increasing water vapour to the final data product.

Only the portion of the curve between 0 and $2.5\,\mathrm{kg\,m^{-2}}$ is of interest in the low regime. For the mid regime, the range of interest is $2 - 6\,\mathrm{kg\,m^{-2}}$, and in the extended regime it is $6 - 12\,\mathrm{kg\,m^{-2}}$. These ranges are different from those given in Table 2 because total water vapour columns are considered here rather than the slant column values. Note however, that the low regime
range for ocean is taken as $0 - 2\,\mathrm{kg\,m^{-2}}$ due to the relatively large deviation in Fig. 7. For MYI, the mid-regime range is taken to be $2 - 4.5\,\mathrm{kg\,m^{-2}}$ due to the low number of measurements beyond that $4.5\,\mathrm{kg\,m^{-2}}$. Also, for the extended regime, only land and ocean measurements are used to determine $\delta b_{23}$. Insignificant variation was seen in the difference of water vapour column between oblique and nadir zenith angles due to a small number of measurements in the extended regime for FYI and MYI surfaces.

The values obtained for $\delta b_{23}$ for different surface types in a given regime are similar. The calculation of $\delta b_{23}$ (Eq. (6)) does not depend on surface properties, and so the same value is expected for all surface types. An average value is therefore used for all surfaces in subsequent retrievals. The values are given in Table 4.

## A.2    Reflectance Ratio Adjustment Parameters

Figure 8 shows the mean total water vapour column differences between oblique (40°–60°) and nadir (0°–20°) satellite local
zenith angles as a function of water vapour column using the same spatial and temporal criteria described previously, with the bias coefficient adjustment parameter having been applied. Each series of curves represents different choices of the reflectance ratio adjustment parameter, $\delta \frac{r_1}{r_2}$. As before, each of four different surface types are considered.

As seen in Fig. 8, there are systematic differences between oblique (40°–60°) and nadir (0°–20°) satellite local zenith angles in the mid regime for a choice of $\delta \frac{r_1}{r_2} = 0$ (i.e., no adjustment). Varying $\delta \frac{r_1}{r_2}$ results in a near-constant offset across a range
of total water vapour column values. The $\delta \frac{r_1}{r_2}$ that minimizes the difference between oblique (40°–60°) and nadir (0°–20°) satellite local zenith angles is chosen.

It is assumed that the correction is necessary because of systematic errors in the retrieval of surface properties. Because $\delta \frac{r_1}{r_2}$ depends on surface type, it is expected the values for each surface will be different.





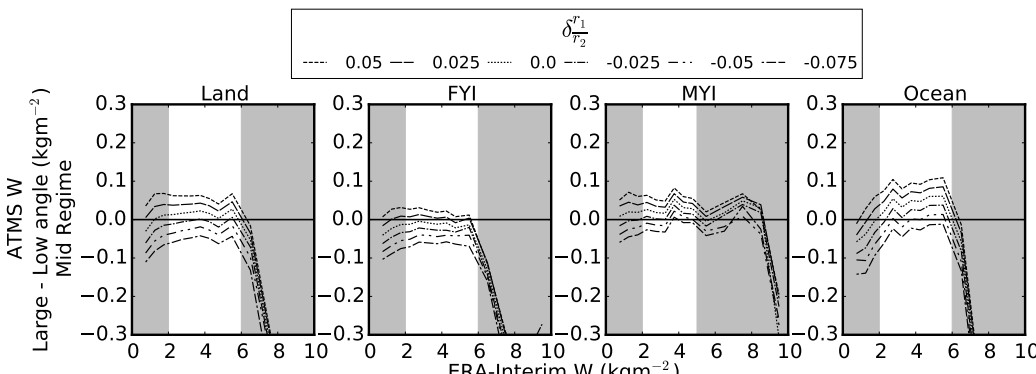

**Figure 8.** Mean differences in the mid regime for ATMS-derived total water vapour columns between oblique ($40°$-$60°$) and nadir ($0°$-$20°$) satellite local zenith angles for different choices of $\delta \frac{r_1}{r_2}$. Gray shading indicates water vapour columns that are outside of the regime of interest.

This method is repeated for determining $\delta \frac{r_1}{r_2}$ in the extended regime. The frequencies for $\delta \frac{r_2}{r_3}$ in the extended regime are the same as those for $\delta \frac{r_1}{r_2}$ in the mid regime, and so $\delta \frac{r_2}{r_3}$ in the extended regime is considered already determined.

The values chosen are given in Table 4. In the mid regime, the reflectivity ratio adjustments are relatively small, with absolute corrections on the order of $0.025$ compared to mean reflectivity ratios of $1.024$ (nadir). It is reasonable to expect reflectivity ratio

errors of this size. The extended regime corrections are somewhat larger, as would be expected given the increased difficulty of measuring reflectivities at extended regime frequencies.

### A.3 Total Water Vapour Column Adjustment Parameter

Figure 9 shows the difference of the mid and low regimes water vapour column for various choices of the water vapour column adjustment parameter ($\delta W$) applied to the mid regime. The value for $\delta W$ is chosen so as to minimize the difference between

regimes in the middle of the region where regimes overlap (see Table 2) for an average satellite local zenith angle of $30°$. Away from this interface point the curves are not flat. This is to be expected from the measurement difficulties that arise when measuring close to the boundary of a regime.

The source of the error requiring this correction is unknown, and will require further study. Individual values are determined for each surface type because of the possibility that the values are surface dependent. The values chosen for each regime

are given in Table 4. Corrections are on the order of $2\%$ in the mid regime and $13\%$ in the extended regime. The mid regime correction is small. The extended regime correction is more substantial, but not surprising given the greater number of potential sources of error for measurements in that regime.





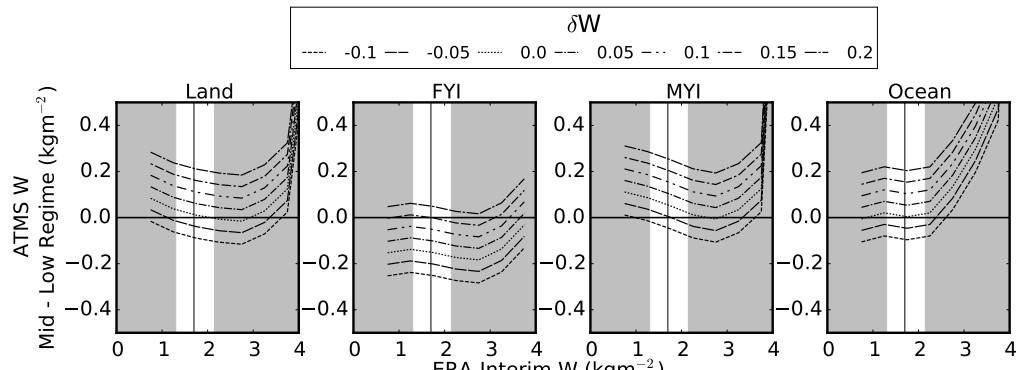

**Figure 9.** Differences in mid and low for the PLDC16 water vapour column at each of the 711 locations combined for different water vapour column adjustments over land, FYI, MYI, and open ocean surfaces compared to ERA-Interim water vapour column. The water vapour column value representing the middle of the overlap between the two regimes for a satellite local zenith angle of $30°$ is given by the dotted line in each plot. Gray shading indicates water vapour columns that are outside the overlap region between the low and mid regimes for measurements with a satellite local zenith angle of $30°$.

*Author contributions.* CP developed the method for the satellite water vapour retrieval, wrote the analysis in Python, performed the data analysis, and prepared the manuscript. This work formed part of CP's doctoral thesis. TJD supervised the doctoral thesis, contributed to the development of the method to retrieve satellite water vapour column, and contributed to the manuscript preparation. GL contributed to the development of the method to retrieve satellite water vapour column and edited the manuscript. KS provided the E-AERI measurements and

edited the manuscript. PMR developed and performed the E-AERI retrievals and edited the manuscript. JRD is the Principal investigator of the PEARL laboratory at which the E-AERI is located and edited the manuscript. RJS is a co-investigator of the PEARL laboratory and edited the manuscript.

*Competing interests.* The authors declare that they have no conflict of interest.

*Acknowledgements.* The authors would like to thank the following for providing resources used in this study: the Center for Satellite Ap-

plications and Research (STAR) Joint Polar Satellite System (JPSS) and the National Oceanic and Atmospheric Administration (NOAA) for ATMS brightness temperature measurements; the Satellite Application Facility for Numerical Weather Prediction (NWP SAF) for the RTTOV radiative transfer model; ECMWF for the ERA-Interim and ERA5 reanalyses; NOAA for the IGRA data set; Japan Meteorological Agency (JMA) for the JRA-55 reanalyses; NOAA/Oceanic and Atmospheric Research (OAR)/Earth System Research Laboratory (ESRL) Physical Sciences Division (PSD) for the NCEP reanalyses; the Atmospheric Radiation Measurement (ARM) user facility, a U.S. Department

of Energy (DOE) Office of Science user facility managed by the Office of Biological and Environmental Research; and ARM GVR mentors, Dr. Maria Cadeddu and Dr. Virendra Ghate, for the GVR data; the Polar Meteorology Group from Ohio State University provided the ASR



V2 data set. We are also grateful for funding from the Natural Sciences and Engineering Research Council (NSERC) and the Canadian Space Agency (CSA) for this research, as well as the Canadian Network for the Detection of Atmospheric Change (CANDAC) for the support of the E-AERI measurements. Penny M. Rowe acknowledges funding from FONDECYT Regular 1161460 and from the National Science Foundation under PLR 1543236.



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



**Table 1.** ATMS instrument specifications including frequencies and nadir polarization orientations. The entry $183.31 \pm 1$ GHz indicastes that dual bands at $183.31 - 1$ GHz and $183.31 + 1$ GHz are combined. Vertical and horizontal polarization refers to cross-track and along-track polarization respectively (Weng and Yang, 2016).

| Frequencies (GHz) | Polarizations |
|---|---|
| 88.2 | Vertical |
| 165.5 | Horizontal |
| $183.31 \pm 1$ | Horizontal |
| $183.31 \pm 1.8$ | Horizontal |
| $183.31 \pm 3$ | Horizontal |
| $183.31 \pm 4.5$ | Horizontal |
| $183.31 \pm 7$ | Horizontal |



**Table 2.** ATMS frequencies for the low, mid, and extended regimes for the retrievals of water vapour column with typical slant water vapour column ($W$) ranges. The frequencies $\nu_1$, $\nu_2$ and $\nu_3$ in each regime are ordered such that the total optical depths $\tau_i$ have $\tau_1 < \tau_2 < \tau_3$.

| Regime | ATMS frequencies (GHz) | | | Slant $W$ (kg m$^{-2}$) |
|---|---|---|---|---|
| | $\nu_1$ | $\nu_2$ | $\nu_3$ | |
| Low | $183.31 \pm 7$ | $183.31 \pm 3$ | $183.31 \pm 1$ | 0 - 2.5 |
| Mid | 165.5 | $183.31 \pm 7$ | $183.31 \pm 3$ | 1.5 - 10 |
| Extended | 88.2 | 165.5 | $183.31 \pm 7$ | 9 - 15 |





**Table 3.** Relative RMS deviations and biases for ATMS-derived water vapour columns compared against columns from reanalyses. Only water vapour columns less than $6\,\mathrm{kg\,m^{-2}}$ were considered. Comparisons with ground based instruments (GVR and E-AERI) and Radiosondes (Russian and Non-Russian Manufactored) are shown as well.

| Dataset/Retrieval | RMS Deviation ($\mathrm{kg\,m^{-2}}$) | Bias ($\mathrm{kg\,m^{-2}}$) |
|---|---|---|
| Instruments | | |
| GVR | 0.42 (15%) | +0.07 (+2.6%) |
| E-AERI | 0.12 (7.2%) | +0.0002 (+0.01%) |
| Russian Manufactored Sondes | 0.69 (21%) | -0.25 (-7.6%) |
| Non Russian Manufactored Sondes | 0.68 (23%) | +0.22 (+7.6%) |
| Reanalyses | | |
| ERA5 | 0.53 (22%) | +0.28 (+11%) |
| ERA-Interim | 0.60 (24%) | +0.24 (+10%) |
| ASR V2 | 0.59 (25%) | +0.31 (+13%) |
| JRA-55 | 0.65 (29%) | +0.51 (+23%) |
| NCEP | 0.93 (40%) | +0.32 (+13%) |





**Table 4.** Summary of corrections applied to ATMS PLDC16 retrievals.

| Surface | Regime | Surface Reflection | $\delta b_{23}$ (K) | $\delta \frac{r_1}{r_2}$ | $\delta W (\mathrm{kg\,m^{-2}})$ |
|---------|--------|--------------------|--------------------|--------------------------|----------------------------------|
| Land | Low | | 1.25 | 0 | 0 |
| | Mid | Lambertian ($S$=0) | 1.75 | -0.025 | 0 |
| | Ext. | | 3.5 | 0.05 | -1.5 |
| FYI | Low | | 1.25 | 0 | 0 |
| | Mid | Lambertian ($S$=0) | 1.75 | 0.025 | 0.15 |
| | Ext. | | 3.5 | 0.15 | -0.9 |
| MYI | Low | | 1.25 | 0 | 0 |
| | Mid | Lambertian ($S$=0) | 1.75 | -0.05 | -0.05 |
| | Ext. | | 3.5 | 0.05 | -1.6 |
| Ocean | Low | | 1.25 | 0 | 0 |
| | Mid | Mixed ($S$=0.5) | 1.75 | -0.05 | 0 |
| | Ext. | | 3.5 | -0.1 | -2.1 |