# Peer review of "Pan-Arctic measurements of wintertime water vapour column using a satellite-borne microwave radiometer"

_Atmospheric Measurement Techniques, 2018_

## Referee Comment (RC2) · Anonymous Referee #1 · 8 Mar 2019

**1    General comments**

The manuscript presents a method for the retrieval of the water vapour column (WVC) in the Arctic using satellite microwave radiometers. It builds on the algorithm published by the same first author a few years ago (Perro et al., 2016). The novelty here is that it uses brightness temperatures measured by a newer instrument (AMTS on Suomi NPP instead of MHS on NOAA-POES and MetOp satellites), and that it takes into account the different reflection and emission properties of the various ground surface types occurring in the Arctic, such as open water, sea ice (first-year ice and multiyear ice) and land. The results of this retrieval for several winter seasons are compared with

ground-based WVC measurements and with meteorological reanalysis data. While the satellite retrieval results compare well with the ground-based measurements, they generally show higher column water vapour than the reanalysis data.

Water vapour is an essential component in most weather and climate related processes. Monitoring water vapour in polar regions is therefore a very relevant topic as such data are sparse, and this study is a useful contribution to this field. There are, however, a number of issues that need clarification, further discussion or analyses. I therefore suggest acceptance after substantial revision.

**2 Specific comments**

1) P.4, L18ff.:
   Here, the authors first introduce the "tuneable parameters" $\delta b_{23}$, $\delta\frac{r_1}{r_2}$, $\delta\frac{r_2}{r_3}$ and $\delta W$. There are several issues here:

   1a) A general one:
       The algorithm presented here (and the one by Perry et al., 2016) is more analytical than the related algorithms by Miao (1999) and Melsheimer and Heygster (2008) because here, the parameters $b_{12}$ and $b_{23}$ are actually calculated using model profiles of the atmosphere, instead of just deriving them empirically from fits with data. The cost for this is, of course, that one needs model or reanalysis data. However, then, the authors still introduce further empirical parameters to adjust the retrieval algorithm. Wouldn't it be easier to determine $b_{12}$ and $b_{23}$ empirically right away, bypassing the need for model/reanalysis data?

   1b) Specifically about $b_{23}$:
       There are actually 3 distinct parameters $b_{23}$, one for each regime, because the numbers 1, 2 and 3 represent different channels in each regime (see

Table 2). To avoid confusion, the parameter names should be different - I suggest a superscript for the regime (L - low, M - mid, X - extended). Therefore, there are also three distinct tuneable parameters $b_{23}$. See also items 9) and 10) further below. The same applies, by the way, for $b_{12}$, but then note that $b_{12}^L = b_{23}^M$ and $b_{12}^M = b_{23}^X$. I suggest to add a small section explaining all this earlier in the manuscript, probably in section 2.3 "Regimes".

1c) Internal calibration?
The authors state that the calibration (determination of the adjustment parameters) "does not depend on outside parameters" (P.5, L.3). I disagree: As we see in Appendix A, all curves used for the parameter determination are plotted with reanalysis WVC values as x-axis.

2) P.5, L.8ff. ("2.2. Surface Reflection Mixtures"):
Reference is made to a still unpublished study of the same first author (Perry et al., 2018, submitted) about the emissivity of the different surface types. This is unfortunate as the main feature that distinguishes the retrieval method in the present manuscript from the method published earlier (Perry et al., 2016), namely, the accounting for varying surface properties, relies on that unpublished study.

3) P.5, L.9/10:
Land is treated as a Lambertian reflector. This is surprising as in the microwave range land is usually treated as specular reflector, unless covered by snow or ice. Do the authors assume here that all land is snow/ice covered? This is probably a reasonable assumption as the study is restricted to the winter months, but this should be mentioned here explicitly.

4) P.6, L.6/7. "... due to the increased retrieval noise with small differences in frequency"
I do not understand this explanation - is the retrieval noise higher for the two channels left out in this study? What do you mean by "small differences in frequency"? The spacing of the sidebands is at 1, 1.8, 3, 4.5 and 7 GHz from the central frequency, the extra channels at 1.8 and 4.5 are not particularly close to the others, at least at first sight. And in which channels are brightness temperatures therefore similar?

5) P.9, L.9/10: "...the range of water vapour values encountered is ... smaller"
- Why are water vapour values in Eureka so much smaller? If this is simply the climatology, that should be briefly mentioned, if not, it should be discussed.

6) P.9, L.11ff. "... sloping terrain", and P.13, L.1-13, and Fig.6:
Why should the topography, or the terrain slope, have an influence on the satellite retrieval or its agreement with ground-based measurements? The physical reasons/mechanisms should be explained and discussed (at least qualitatively). Is it just the effect of the "shorter" air column above elevated ground? But are the elevation variations near the measurement stations large enough to cause the observed effect?

7) Sections 4 (Radiosonde Comparison) and 5 (Reanalyses Comparison):
The algorithm, using the three regimes, can retrieve up to 14 kg/m2 WVC. In all the comparisons, the authors take into account retrieved values up to only 6 kg/m2 (low and mid range regimes only) - the reason or motivation for this is not given. This should be explained and discussed, or else the whole range should be used. (Note also that the WVC range shown in the plot in Fig. 4 is actually 0 to 10 kg/m2, although RMSD and bias are calculated only for WVC < 6 kg/m2, which is confusing)

8) P.15, L.2-4:
Why are oblique measurements drier than nadir measurements? Is there a physical reason for that? Maybe some saturation effect? This should be discussed.

9) P.15, L.6-8, about the adjustment parameters:

As mentioned above in item 1b), the authors must state clearly that there are three separate adjustment parameters $\delta b_{23}$, one for each regime (see above the suggestion with the superscripts).

10) P.15. Section A.1 ("Bias Coefficient Adjustment Parameters")
The authors should state more clearly that they show a plot for the determination of one of the three parameters only, or they should rather state that the adjustment parameters for the mid and extended regime have been determined in a similar way.

11) The effect of clouds has been neglected in this study. However, in particular ice clouds have a strong effect on the 183 GHz channels because of scattering. These channels are even used for the detection of strong convection associated with, e.g., polar lows. The effect of ice clouds on this kind of algorithm are erroneously low water vapour retrievals (see, e.g., doi:10.1109/JSTARS.2015.2499083)

**3  Technical corrections**

P.8, L.4: constaint -> constraint

P.8, L.14 and P.12, caption of Fig. 5, and elsewhere: "comparator data product": 'comparator' is the wrong word here. Rather "data product with which the comparison was done"

Fig.3 and Fig.5: Maybe express the relative RMS deviation and relative bias in per cent. If plain number are given, they might be misunderstood as per cent and will be much too low. In addition, in the text, the authors use per cent.

**4  Acronyms used here**

WVC: water vapour column

---

## Author Comment (AC1) · 1 Apr 2019

We thank Reviewer 1 for their constructive comments, and agree that a better understanding of polar water vapour is important. While the revisions proposed were deemed as "substantial", we think that they can be addressed in the time frame allowed by this review. We note that Reviewer 2 raised some of the same points.

Rather than respond point by point at this time, we will address a selection of issues in what we view as their order of significance. Remaining issues and revision details will be provided in our final response to the review after discussion has closed.

[Figure]

1) Reviewer #1: "The effect of clouds has been neglected in this study."

Authors: We agree that there is a potential cloud effect, and that it should be discussed. The impact of clouds on water vapour retrievals is discussed in detail by C. Perro in his Ph.D. thesis (Dalhousie University, 2017, Chapter 3, https://dalspace.library.dal.ca/handle/10222/73353). He found that the statistical impact of clouds on comparisons between radiosonde and satellite water vapour columns was small. This point was discussed briefly in Perro et al. (2016). We are presently revisiting this analysis, with a view toward publication.

2) Reviewer #1: "... the authors take into account retrieved values up to only 6 kg/m2 (low and mid range regimes only) - the reason or motivation for this is not given."

We find that the extended regime retrievals are noisier, and the tuneable correction parameters are larger. The noise in the extended regime would unduly affect the statistics of the low and mid regimes. Limiting our analysis to 6 kg/m2 and below eliminates these problems and is consistent with the earlier analysis of Perro et al. (2016).

3) Reviewer #1: "Reference is made to a still unpublished study of the same first author..."

Authors: This paper is still under review, and represents an updated analysis of that given in the thesis of C. Perro. We expect to post the paper on archive.org for the next stage of review.

4) Reviewer #1: "Do the authors assume here that all land is snow/ice covered?"

Authors: As a first analysis, we have divided the surface into four categories: first-year ice, multi-year ice, land and ocean. We agree that subdividing land into further categories would be useful, and expect to pursue that in future work. It seems reasonable to assume that most land surfaces in this winter study are snow covered.

In the thesis of C. Perro, the surface reflection assumptions were tested against actual water vapour retrievals. The same result for land was obtained as was found in the

above-mentioned submitted paper. We propose to include these results in an appendix to a revised manuscript.

5) Reviewer #1: "Wouldn't it be easier to determine b12 and b23 empirically right away, bypassing the need for model/reanalysis data?"

Authors: The approach advocated is effectively the same as that of Melsheimer and Heygster (2008). As discussed by Perro et al. (2016), our new approach is more accurate at the cost of computational complexity. The tuneable parameters create consistency between and within regimes, without removing the advantages of computing bias coefficients from auxiliary profiles that vary in time and space.

6) Reviewer #1: "1c) Internal calibration? The authors state that the calibration (determination of the adjustment parameters) "does not depend on outside parameters" (P.5, L.3). I disagree: As we see in Appendix A, all curves used for the parameter determination are plotted with reanalysis WVC values as x-axis."

Authors: The point we are trying to make is that the data are not calibrated to match those of any external source. For example, we calibrate oblique satellite measurements against nadir satellite measurements, but not against radiosondes or other external measurements. We will find a better way to express this in the revised manuscript.

---

## Referee Comment (RC3) · Anonymous Referee #2 · 3 Apr 2019

Review of "Pan-Arctic measurements of wintertime water vapour column using a satellite-borne microwave radiometer" by C. Perro et al.

General comments:

This study builds upon a previous AMT paper's water vapour retrieval by adding further validation/comparison, improved reflectances, and some tunable parameters into the equation relating TBs, reflectances, and transmittances. The authors implicitly argue that the retrieval is to be trusted on virtue of the good validation against two ground-based sensors, which in turn means that all the reanalyses examined might have a systematic bias.

While it is an interesting study and well written, there are too many pieces missing here for it to be recommended for publication. The methodology is quite light, relying heavily on the Perro et al. (2016) study and a submitted manuscript to IEEE TGRS (therefore not reviewable at this time) on the reflectances methods that is one of the key novelties of this updated study. It is also disconcerting that an undefined tuning parameter is introduced at the end of the appendix (delta W) but described in no detail. In addition there is no discussion of the retrieval's error sources or uncertainties, including the influence of supercooled cloud water or ice clouds, which can certainly impact TBs at these frequencies.

The study's conclusions are fittingly light as well, but few of these are significant because of the study's limited scope. Why are all comparisons limited to 6kg of water vapour or less? Because all comparisons are from this subset, it is hard to judge the retrieval's performance for different conditions. And if the retrieved quantities do not permit strong conclusions, then assessment of the radiative transfer underpinning the retrieval could yield more significant conclusions. But instead the authors shy away from giving explanations for some of these aspects, such as the possible cause for elevation changes causing errors or why such systematic errors exist that necessitate the tuning parameters outlined in the appendix.

My recommendation is to reject the current manuscript but encourage resubmission once the study has been fleshed out by addressing the following major criticisms. A selection of minor/textual comments follows the major points.

Major:

1. It is unacceptable to rely upon unpublished work as a foundational part of the retrieval methodology, and it is not correct to list a submitted manuscript as 'Perro et al. (2018)' in the text when it is not published as this is misleading. It is difficult for authors when coincident manuscripts are in the review process, but it means that a reviewer cannot judge the methods fully because the information is just not available. If this
were a very minor part of the methodology then it would not be as big a deal, but it is cited numerous times.

2. A tuning parameter, delta W, is confusingly introduced at the end of the appendix. It is not defined in any equation that I saw, and then it's admitted that the "source of the error requiring this correction is unknown" (P17 L13). A quick glance at Table 4 shows that this parameter can be as big as 2kg? It's unacceptable to have this separate from the main methodology when its magnitude is so large. It would also be good to have some discussion of the magnitude of the other tuning parameters in Table 4 and what these mean; differences in the bias coefficients and reflectances of up to 3.5K and 0.15 are quite significant indeed, but yet gets no discussion. To me these represent large enough tuning to signal the inadequacy of the forward model in some regimes.

3. Clouds are only mentioned in the manuscript when referring to the IR instrument, as it only performs retrieval in clear-sky. They are not treated in the forward model, which would be fine if they were entirely radiatively insignificant, but both liquid and ice clouds have a non-negligible impact on microwave radiances. Even in the Arctic wintertime, supercooled water can exist in clouds (Cesana et al. 2012 (https://doi.org/10.1029/2012GL053385)), and thicker ice clouds can exert a significant scattering signal on radiances near 183GHz. If this were admitted as an error source and quantified that would be better, but instead it seems these errors are just folded into the myriad tuning parameters of Table 4.

4. All of the comparison and validation is limited to cases of <6kg water vapour. This seems arbitrary, and was never justified. Why are higher values not included, or the results stratified? This was especially confusing because of the separation of the forward model parameters into the low/mid/extended water vapour regimes.

5. The validation against E-AERI left a few unanswered questions. Firstly, the bias as reported in Table 3 of '+0.0002' should not be written that way as surely the measurement accuracy of either retrieval does not approach that many significant figures.

Secondly, the zero bias and very low RMS error seems unlikely, given the different fields of view, sensor noise, and imperfect space-time matchups. Do the E-AERI retrievals within some time window, let alone over some spatial domain, vary more or less than the reported RMS deviation of 0.12kg? Some context here would be very useful for readers to judge how good this result is, and more importantly its statistical significance. Because there is no discussion of retrieval errors, or possible errors from E-AERI itself, it is hard to judge such questions.

6. Inclusion of Figure 6 does not seem to be justified. It is a speculative assertion that is not underpinned by any statistics (say correlation between the two) and there is no physical mechanism hypothesized to be behind it (P13 L18), so its inclusion caused more confusion than insight.

Minor/textual issues:

P2 L25: This paragraph contains results of a kind, comparing ERA5 to other reanalyses, so it seems inappropriate in the introduction. Further, none of the reanalysis acronyms are defined.

P5 L7: It's speculative to say that these 'are not expected to impact the conclusions' so this should be removed.

P5 L27: "A formula was fit" is vague and should be explained.

P6 L7: What does "increased retrieval noise" mean here?

P7 L5: Why would more water vapour over the ocean cause reflectance ratio errors? This requires physical explanation.

P8 L14: I don't think this is the correct use of "comparator" though it may be a usage I am unfamiliar with.

P8 L16: Improved relative to what?

P10 Fig3: There should be units given for A and B.

P10 L1: "scientists" should be "sondes"?

P11 L3: What kind of "regular grid" was chosen? And are the 711 locations all of the points on this grid or were they chosen in some way?

P14 L9: There should be at least one citation to back this up.

---

## Author Comment (AC2) · 3 Jun 2019

**1   Response to Referee 1:**

**1 General comments**

**Referee 1:**   The manuscript presents a method for the retrieval of the water vapour column (WVC) in the Arctic using satellite microwave radiometers. It builds on the algorithm published by the same first author a few years ago (Perro et al., 2016). The novelty here is that it uses brightness temperatures measured by a newer instrument (AMTS on Suomi NPP instead of MHS on NOAA-POES and MetOp satellites), and

that it takes into account the different reflection and emission properties of the various ground surface types occurring in the Arctic, such as open water, sea ice (first-year ice and multiyear ice) and land. The results of this retrieval for several winter seasons are compared with ground-based WVC measurements and with meteorological reanalysis data. While the satellite retrieval results compare well with the ground-based measurements, they generally show higher column water vapour than the reanalysis data.

Water vapour is an essential component in most weather and climate related processes. Monitoring water vapour in polar regions is therefore a very relevant topic as such data are sparse, and this study is a useful contribution to this field. There are, however, a number of issues that need clarification, further discussion or analyses. I therefore suggest acceptance after substantial revision.

**Authors:** Thank you for your comments. We agree that more attention needs to be paid to Arctic water vapour, and appreciate your recommendation to publish after substantial revision. We take your concerns seriously and have addressed all of the points your raised in your review.

**2 Specific comments**

**Referee 1:** 1) P.4, L18ff.: Here, the authors first introduce the "tuneable parameters" $\delta$b23, $\delta$r1r2 , $\delta$r2r3 and $\delta$W. There are several issues here:

1a) A general one: The algorithm presented here (and the one by Perry et al., 2016) is more analytical than the related algorithms by Miao (1999) and Melsheimer and Heygster (2008) because here, the parameters b12 and b23 are actually calculated using model profiles of the atmosphere, instead of just deriving them empirically from fits with data. The cost for this is, of course, that one needs model or reanalysis data.

However, then, the authors still introduce further empirical parameters to adjust the retrieval algorithm. Wouldn't it be easier to determine b12 and b23 empirically right away, bypassing the need for model/reanalysis data?

**Authors:** The approach advocated above is effectively the same as that of Melsheimer and Heygster (2008), which has fixed values for the bias coefficients $b_{12}$ and $b_{23}$ for a given viewing angle. Our bias coefficients vary in time and space with the atmospheric conditions. As discussed by Perro et al. (2016), our new approach is more accurate at the cost of computational complexity. We will clarify these points in Sec. 2.1 of the revised manuscript.

In our analysis we find that there is a dependence on viewing angle and discontinuities between regimes. These issues are eliminated in Melsheimer and Heyster's (2008) analysis by calibrating their retrievals against radiosonde measurements. Because we wish for our retrieval to be independent of radiosonde measurements, we have introduced correction factors to allow for an internal calibration. Effectively, the correction factors force measurements at different viewing angles to agree. This point will also be clarified in the revised manuscript.

**Referee 1:** 1b) Specifically about b23: There are actually 3 distinct parameters b23, one for each regime, because the numbers 1, 2 and 3 represent different channels in each regime (see Table 2). To avoid confusion, the parameter names should be different - I suggest a superscript for the regime (L - low, M - mid, X - extended). Therefore, there are also three distinct tuneable parameters b23. See also items 9) and 10) further below. The same applies, by the way, for b12, but then note that bL12 = bM23 and bM12 = bX23. I suggest to add a small section explaining all this earlier in the manuscript, probably in section 2.3 "Regimes".

**Authors:** Thank you for this suggestion, which will indeed help us to better explain a complex issue. This will also make our work more consistent with MH2008's since they use a similar notation to what is suggested.

**Referee 1:** 1c) Internal calibration? The authors state that the calibration (determination of the adjustment parameters) "does not depend on outside parameters" (P.5, L.3). I disagree: As we see in Appendix A, all curves used for the parameter determination are plotted with reanalysis WVC values as x-axis.

**Authors:** The point we are trying to make is that the data are not calibrated to match those of any external source. For example, we calibrate our oblique satellite measurements against our nadir satellite measurements, but not against radiosondes or other external measurements. We will clarify this point in the revised manuscript.

**Referee 1:** 2) P.5, L.8ff. ("2.2. Surface Reflection Mixtures"): Reference is made to a still unpublished study of the same first author (Perry et al., 2018, submitted) about the emissivity of the different surface types. This is unfortunate as the main feature that distinguishes the retrieval method in the present manuscript from the method published earlier (Perry et al., 2016), namely, the accounting for varying surface properties, relies on that unpublished study.

**Authors:** This paper is under review, and represents an updated analysis of that given in the Ph. D. thesis by Perro (2017; https://dalspace.library.dal.ca/handle/10222/73353). We have emailed a pdf of the draft paper to the editor to share with both Referees.

While it would have been better to have had the surface emissivity paper published prior to submitting this paper on water vapour measurements, this was not a practical

possibility given the time, funding, and career development constraints on a PostDoc-toral fellow (Perro).

**Referee 1:** 3) P.5, L.9/10: Land is treated as a Lambertian reflector. This is surprising as in the microwave range land is usually treated as specular reflector, unless covered by snow or ice. Do the authors assume here that all land is snow/ice covered? This is probably a reasonable assumption as the study is restricted to the winter months, but this should be mentioned here explicitly.

**Authors:** For the present analysis, we divided the surface into four categories: first-year ice, multi-year ice, land and ocean. Subdividing land into further categories would be useful, and expect to pursue that in future work. It seems reasonable to assume that most land surfaces in this winter study are snow covered, and we will state that explicitly in the revised manuscript.

We presented evidence in the surface emissivity paper currently under review that shows convincingly that land should be treated as a Lambertian reflector. We will emphasize this point, and that it is different from what is typically done, in the revised manuscript.

**Referee 1:** 4) P.6, L.6/7. "... due to the increased retrieval noise with small differences in frequency" I do not understand this explanation - is the retrieval noise higher for the two channels left out in this study? What do you mean by "small differences in frequency"? The spacing of the sidebands is at 1, 1.8, 3, 4.5 and 7 GHz from the central frequency, the extra channels at 1.8 and 4.5 are not particularly close to the others, at least at first sight. And in which channels are brightness temperatures therefore similar?

**Authors:**

Equation 6, which forms the basis for the water vapour retrieval in our work (and that of Miao et al. (2001) and Melsheimer and Heygster (2008)), require brightness temperature difference measurements. Small differences in temperature lead to large relative errors. We therefore excluded certain channels from our analysis to avoid this difficulty. All immediately adjacent 183 GHz channels have similar brightness temperatures. We will replace the confusing sentence in our original manuscript with an explanation based on this discussion.

**Referee 1:** 5) P.9, L.9/10: "...the range of water vapour values encountered is ... smaller" - Why are water vapour values in Eureka so much smaller? If this is simply the climatology, that should be briefly mentioned, if not, it should be discussed.

**Authors:** Yes, the smaller range of water vapour values at Eureka is climatological. We will indicate this in the updated manuscript and refer to the climatology at Eureka of Lesins et al. (Atmosphere-Ocean, 2010, https://www.tandfonline.com/doi/abs/10.3137/AO1103.2010) for support.

**Referee 1:** 6) P.9, L.11ff. "... sloping terrain", and P.13, L.1-13, and Fig.6: Why should the topography, or the terrain slope, have an influence on the satellite retrieval or its agreement with ground-based measurements? The physical reasons/mechanisms should be explained and discussed (at least qualitatively). Is it just the effect of the "shorter" air column above elevated ground? But are the elevation variations near the measurement stations large enough to cause the observed effect?

**Authors:** Referee 2 also remarked on this section, and found it too speculative. We agree with their assessment and have decided to remove it from the paper.

In answers to your questions, you are correct, the shorter air column is the effect we considered that could influence the retrieved water vapour column. The effect is not large enough to significantly influence our comparison.

**Referee 1:** 7) Sections 4 (Radiosonde Comparison) and 5 (Reanalyses Comparison): The algorithm, using the three regimes, can retrieve up to 14 kg/m2 WVC. In all the comparisons, the authors take into account retrieved values up to only 6 kg/m2 (low and mid range regimes only) - the reason or motivation for this is not given. This should be explained and discussed, or else the whole range should be used. (Note also that the WVC range shown in the plot in Fig. 4 is actually 0 to 10 kg/m2, although RMSD and bias are calculated only for WVC < 6 kg/m2, which is confusing)

**Authors:** We find that the extended regime retrievals are noisier, and the tuneable correction parameters are larger. The noise in the extended regime would unduly affect the statistics of the low and mid regimes. Limiting our analysis to 6 kg/m2 and below eliminates these problems and is consistent with the earlier analysis of Perro et al. (2016). We will address this in the revised manuscript, and provide statistics to indicate what percentage of the measurements are eliminated by this choice.

We believe it is still valuable to plot measurements up to 10 kg/m2, because it makes readily apparent the larger error at higher values and that most columns measured are below 6 kg/m2. The plot also shows that the trend of relative dryness in ECMWF extends to the higher water vapour columns. We will make this point in the revised manuscript.

**Referee 1:** 8) P.15, L.2-4: Why are oblique measurements drier than nadir measurements? Is there a physical reason for that? Maybe some saturation effect? This should

be discussed.

**Authors:**

We have performed simulations to test the saturation hypothesis, and found it could not explain relatively dry retrievals at oblique angles. We will make this point in the revised manuscript.

There are a few possible reasons that could introduce the errors we are seeing:

1) Issue with the calibration of the satellite instrument brightness temperatures.

2) Auxiliary profile information having a consistent bias in all or part of the temperature or water vapour shape profiles.

3) Errors in the RTTOV radiative transfer scheme.

To determine which of these is causing the issue with the retrievals is difficult – we must rely on the output of other groups in this work – and so we determined it best to use the empirical corrections included in this manuscript. We will address these points in the revised manuscript.

**Referee 1:** 9) P.15, L.6-8, about the adjustment parameters: As mentioned above in item 1b), the authors must state clearly that there are three separate adjustment parameters $\delta$b23, one for each regime (see above the suggestion with the superscripts).

**Authors:** Agreed. We will make this recommended change to the revised manuscript.

**Referee 1:** 10) P.15. Section A.1 ("Bias Coefficient Adjustment Parameters") The authors should state more clearly that they show a plot for the determination of one of the three parameters only, or they should rather state that the adjustment parameters

for the mid and extended regime have been determined in a similar way.

**Authors:** Agreed. We will include both points in the revised manuscript.

**Referee 1:** 11) The effect of clouds has been neglected in this study. However, in particular ice clouds have a strong effect on the 183 GHz channels because of scattering. These channels are even used for the detection of strong convection associated with, e.g., polar lows. The effect of ice clouds on this kind of algorithm are erroneously low water vapour retrievals (see, e.g., doi:10.1109/JSTARS.2015.2499083)

**Authors:** We agree that there is a potential cloud effect, and that it should be discussed. The impact of clouds on water vapour retrievals was considered by Perro (Ph. D. thesis, 2017). He found that the statistical impact of clouds on comparisons between radiosonde and satellite water vapour columns was small. This point was discussed briefly in Perro et al. (2016).

The paper provided by the referee is helpful in assessing the potential impact on our conclusions. Because clouds are thought to cause a dry bias in microwave water vapour retrievals, they cannot explain the fact that our measurements are relatively moist compared to ERA5. We will make this point in our revised manuscript.

A detailed analysis of how clouds impact our retrieval will require a separate publication. The analysis would be too extensive to include here. Fully assessing the impact of clouds is in our near-term plans.

**3 Technical corrections**

**Referee 1:** P.8, L.4: constaint -> constraint

**Authors:** We will make this change.

**Referee 1:** P.8, L.14 and P.12, caption of Fig. 5, and elsewhere: "comparator data product": 'comparator' is the wrong word here. Rather "data product with which the comparison was done"

**Authors:** We will make this change.

**Referee 1:** Fig.3 and Fig.5: Maybe express the relative RMS deviation and relative bias in per cent. If plain number are given, they might be misunderstood as per cent and will be much too low. In addition, in the text, the authors use per cent.

**Authors:** We will make this change.

---

## Author Comment (AC3) · 3 Jun 2019

**1   Response to Referee 2:**

**General comments:**

**Referee 2:**   This study builds upon a previous AMT paper's water vapour retrieval by adding further validation/comparison, improved reflectances, and some tunable parameters into the equation relating TBs, reflectances, and transmittances. The authors implicitly argue that the retrieval is to be trusted on virtue of the good validation against two groundbased sensors, which in turn means that all the reanalyses examined might

have a systematic bias.

While it is an interesting study and well written, there are too many pieces missing here for it to be recommended for publication. The methodology is quite light, relying heavily on the Perro et al. (2016) study and a submitted manuscript to IEEE TGRS (therefore not reviewable at this time) on the reflectances methods that is one of the key novelties of this updated study. It is also disconcerting that an undefined tuning parameter is introduced at the end of the appendix (delta W) but described in no detail. In addition there is no discussion of the retrieval's error sources or uncertainties, including the influence of supercooled cloud water or ice clouds, which can certainly impact TBs at these frequencies.

The study's conclusions are fittingly light as well, but few of these are significant because of the study's limited scope. Why are all comparisons limited to 6kg of water vapour or less? Because all comparisons are from this subset, it is hard to judge the retrieval's performance for different conditions. And if the retrieved quantities do not permit strong conclusions, then assessment of the radiative transfer underpinning the retrieval could yield more significant conclusions. But instead the authors shy away from giving explanations for some of these aspects, such as the possible cause for elevation changes causing errors or why such systematic errors exist that necessitate the tuning parameters outlined in the appendix.

My recommendation is to reject the current manuscript but encourage resubmission once the study has been fleshed out by addressing the following major criticisms. A selection of minor/textual comments follows the major points.

**Authors:** Thank you for your feedback on our manuscript. We will respond to each of the above points below, where they are repeated in more detail. We believe that we have a good handle on each of the major points owing to the work put into the previously published Ph.D. thesis by Perro (2017; https://dalspace.library.dal.ca/handle/10222/73353). We therefore hope that the Referee will be open to considering a revised manuscript.

**Major:**

**Referee 2:** 1. It is unacceptable to rely upon unpublished work as a foundational part of the retrieval methodology, and it is not correct to list a submitted manuscript as 'Perro et al. (2018)' in the text when it is not published as this is misleading. It is difficult for authors when coincident manuscripts are in the review process, but it means that a reviewer cannot judge the methods fully because the information is just not available. If this were a very minor part of the methodology then it would not be as big a deal, but it is cited numerous times.

**Authors:** The paper by Perro et al. (2018) remains under review, and represents an updated analysis of that given in the Ph. D. thesis by Perro (2017; https://dalspace.library.dal.ca/handle/10222/73353). We have emailed a pdf of the draft paper to the editor to share with both Referees.

While it would have been better to have had the surface emissivity paper published prior to submitting this paper on water vapour measurements, this was not a practical possibility given the time, funding, and career development constraints on a PostDoctoral fellow (Perro).

**Referee 2:** 2. A tuning parameter, delta W, is confusingly introduced at the end of the appendix. It is not defined in any equation that I saw, and then it's admitted that the "source of the error requiring this correction is unknown" (P17 L13). A quick glance at Table 4 shows that this parameter can be as big as 2kg? It's unacceptable to have this separate from the main methodology when its magnitude is so large.

**Authors:** It is not possible to include delta W in an equation because the water vapour column is an implicit parameter of Eq. 9 and determined using an optimization
procedure. The delta W correction of 2 kg/m2 is for the extended regime only. The choice of limiting our comparisons to columns less than 6 kg/m2 means that only the low and mid regimes – where the corrections are small – are used. We agree that this is not obvious and will add clarifying points to the revised manuscript.

Despite much effort, we still do not know why the delta W correction for the extended regime is so large. We made that point clear in the manuscript in the interests of transparency.

**Referee 2:** It would also be good to have some discussion of the magnitude of the other tuning parameters in Table 4 and what these mean; differences in the bias coefficients and reflectances of up to 3.5K and 0.15 are quite significant indeed, but yet gets no discussion. To me these represent large enough tuning to signal the inadequacy of the forward model in some regimes.

**Authors:** The numbers quoted above are for the extended regime, which is not included in the comparative work in this manuscript. We agree that there may be some inadequacy in the forward model (RTTOV, the radiative transfer model published and used by ECMWF). To this we would add the following additional sources of error:

1) Issue with the calibration of the satellite instrument brightness temperatures.

2) Auxiliary profile information having a consistent bias in all or part of the temperature or water vapour shape profiles.

We will make these points in the revised manuscript. To determine which of these is causing the issue with the retrievals is difficult – we must rely on the output of other groups in this work – and so we determined it best to use the empirical corrections included in this manuscript.

**Referee 2:** Clouds are only mentioned in the manuscript when referring to the IR instrument, as it only performs retrieval in clear-sky. They are not treated in the forward model, which would be fine if they were entirely radiatively insignificant, but both liquid and ice clouds have a non-negligible impact on microwave radiances. Even in the Arctic wintertime, supercooled water can exist in clouds (Cesana et al. 2012 (https://doi.org/10.1029/2012GL053385)), and thicker ice clouds can exert a significant scattering signal on radiances near 183GHz. If this were admitted as an error source and quantified that would be better, but instead it seems these errors are just folded into the myriad tuning parameters of Table 4.

**Authors:** We agree that there is a potential cloud effect, and that it should be discussed. In particular, we will refer to the paper of Cesana et al. (2012).

The impact of clouds on water vapour retrievals was considered by Perro (Ph. D. thesis, 2017). He found that the statistical impact of clouds on comparisons between radiosonde and satellite water vapour columns was small. This point was discussed briefly in Perro et al. (2016).

A paper suggested by Referee 1 (doi:10.1109/JSTARS.2015.2499083) is helpful in assessing the potential impact on our conclusions. Because clouds are thought to cause a dry bias in microwave water vapour retrievals, they cannot explain the fact that our measurements are relatively moist compared to ERA5. We will make this point in our revised manuscript.

A detailed analysis of how clouds impact our retrieval will require a separate publication. The analysis would be too extensive to include here. Fully assessing the impact of clouds is in our near-term plans.

**Referee 2:** 4. All of the comparison and validation is limited to cases of <6kg water vapour. This seems arbitrary, and was never justified. Why are higher values

not included, or the results stratified? This was especially confusing because of the separation of the forward model parameters into the low/mid/extended water vapour regimes.

**Authors:** We agree that the reasons for this choice need to be made clear, and will do so in the revised manuscript. We find that the extended regime retrievals are noisier, and the tuneable correction parameters are larger, than in either the low or mid regimes. We therefore have less confidence in the extended regime data. Limiting our analysis to 6 kg/m2 and below eliminates the extended regime from comparisons and is consistent with the earlier analysis of Perro et al. (2016).

**Referee 2:** 5. The validation against E-AERI left a few unanswered questions. Firstly, the bias as reported in Table 3 of '+0.0002' should not be written that way as surely the measurement accuracy of either retrieval does not approach that many significant figures. Secondly, the zero bias and very low RMS error seems unlikely, given the different fields of view, sensor noise, and imperfect space-time matchups. Do the E-AERI retrievals within some time window, let alone over some spatial domain, vary more or less than the reported RMS deviation of 0.12kg? Some context here would be very useful for readers to judge how good this result is, and more importantly its statistical significance. Because there is no discussion of retrieval errors, or possible errors from E-AERI itself, it is hard to judge such questions.

**Authors:**

We appreciate the concern here: the comparison between the two data sets indeed looks very good, but there is no uncertainty associated with the measured bias. We will obtain an uncertainty estimate by determining the bias implied by errors in linear least-squares fits to our data.

We note that the E-AERI data in our paper represented 7-minute integrations. Assum-

ing an average wind speed of 20 m/s leads to an equivalent spatial resolution of about 8.4 km. The ATMS has a spatial resolution of 15 km at nadir. Thus, the resolutions of the two instruments are reasonably compatible. Higher resolution E-AERI data are not available, and so it is not possible for us to look at variations in a smaller time window. Variations in a longer time window would need to be compared to larger spatial integrations of the ATMS measurements.

Errors in the E-AERI operating at Eureka and in ancillary data used in the retrievals were characterized and estimated as described in Knuteson et al (2004a,b), Rowe et al (2008) and Weaver et al (2017). These errors were propagated to uncertainties in retrieved water vapour (Weaver et al 2017). In previous work (Weaver et al 2017), uncertainties were found to be 3 to 11% for summer to winter at Eureka and water vapour retrievals from the E-AERI were found to agree with several other instruments to within combined measurement errors, with large biases (1 to 3%) and comparably small standard deviations (0.1 to 0.3%) between instruments. For this work, uncertainties in retrieved water vapour are typically 8 to 11% for December-March cases at Eureka.

We will address the above issues in the revised manuscript.

**Referee 2:** 6. Inclusion of Figure 6 does not seem to be justified. It is a speculative assertion that is not underpinned by any statistics (say correlation between the two) and there is no physical mechanism hypothesized to be behind it (P13 L18), so its inclusion caused more confusion than insight.

**Authors:** We will remove figure 6 and the discussion surrounding the sloping terrain.

**Minor/textual issues:**

**Referee 2:** P2 L25: This paragraph contains results of a kind, comparing ERA5 to other reanalyses, so it seems inappropriate in the introduction. Further, none of the reanalysis acronyms are defined.

**Authors:** We will remove this paragraph and will expand the acronyms for the various data products (and provide references as appropriate) where they are first introduced.

**Referee 2:** P5 L7: It's speculative to say that these 'are not expected to impact the conclusions' so this should be removed.

**Authors:** We will remove this statement, as requested.

**Referee 2:** P5 L27: "A formula was fit" is vague and should be explained.

**Authors:** We can explain this more simply by saying that the intermediate values were interpolated to higher resolution.

**Referee 2:** P6 L7: What does "increased retrieval noise" mean here?

**Authors:**

This issue was also raised by Referee 1. Equation 6, which forms the basis for the water vapour retrieval in our work (and that of Miao et al. (2001) and Melsheimer and Heyster (2008)), require brightness temperature difference measurements. Small differences in temperature lead to large relative errors – what we called "retreival noise" in the original manuscript. All immediately adjacent 183 GHz channels have similar brightness temperatures. We therefore excluded certain channels from our analysis to

avoid this difficulty. We will replace the confusing sentence in our original manuscript with an explanation based on this discussion.

**Referee 2:** P7 L5: Why would more water vapour over the ocean cause reflectance ratio errors? This requires physical explanation.

**Authors:** Measured radiances have components due to both the surface and atmospheric water vapour. Removal of the atmospheric water vapour contribution using auxiliary data is more uncertain (owing to errors in the auxiliary data) when the water vapour columns are greater, as is climatologically the case over ocean. We will provide this physical explanation in the revised manuscript.

**Referee 2:** P8 L14: I don't think this is the correct use of "comparator" though it may be a usage I am unfamiliar with.

**Authors:** Referee 1 also raised this point. We will replace "comparator data product" with "data product being compared".

**Referee 2:** P8 L16: Improved relative to what?

**Authors:** We mean improved relative to our 2016 analysis, and will make that clear in the revised manuscript.

**Referee 2:** P10 Fig3: There should be units given for A and B.

**Authors:** The change is a relative change so A and B are unitess. In the revised manuscript we will represent these as percentages so that the units are clear.

**Referee 2:** P10 L1: "scientists" should be "sondes"?

**Authors:** Agreed. We will fix this typo.

**Referee 2:** P11 L3: What kind of "regular grid" was chosen? And are the 711 locations all of the points on this grid or were they chosen in some way?

**Authors:** The latitudinal grid resolution is constant at 2.5 degrees. Below 80 N the longitudinal grid resolution is 5 degrees, between 80 and 85 N it is 10 degrees, and above 85 N it is 20 degrees. Yes, the 711 locations represent all of the points on the grid. We will clarify these points in the revised manuscript.

**Referee 2:** P14 L9: There should be at least one citation to back this up.

**Authors:** We will cite Scarlat et al. (2018) and Melsheimer and Heygster (2008).

---

## Editor Comment (EC1) · Patrick Eriksson (Editor) · 25 Jun 2019

Dear Christopher Perro and co-authors,

After a new reading of the manuscript, I have decided to follow the advice of referee 2 to reject the manuscript. Referee 1 raises similar concerns and suggests a major revision. That is, it is clear that there are important issues to fix in the manuscript and I find myself to agree more with referee 2 in the weight to put on the issues. In addition, I don't see any clear suggestions in the replies to the referees how the main problems shall be removed. When it comes to details, I mainly refer to the referee reports but I will make some comments.

[Figure]

A main problem is that this manuscript relies strongly on another submitted manuscript. You have now emailed me this manuscript, but it can not be expected that the referees of this manuscript also shall review that one. I don't see any other solution than to wait until the other review is finished. In this manuscript, the other manuscript is referred to as Perro et al (2018) and I missed in my initial reading that it was only submitted. If I have noticed that, I would have rejected this manuscript at submission.

The links to the other manuscript are especially problematic as the retrievals presented here show a clear dependence to incidence angle. In response to the referees, some reasons are suggested, but I don't find them highly convincing. At least, I find it as likely that the angle-dependent bias originates in the treatment of the surface, that makes it critical to see if the results in the other manuscript will pass review or not.

I understand that perfect, final retrievals cannot be expected at this stage. On the other hand, there can not be too many articles on the way to a final version. Anyhow, it is hard to judge what progress this manuscript actually provides, when large correction factors are still needed and several critical issues are left for future studies (such as the impact of clouds).

Some extra remarks, not raised by the referees:

* Two stations provide a weak basis for validation. You can also use GRUAN stations.

* Today with fast computers, I don't understand why Eq 1 is used instead of a full treatment of the radiative transfer.

* At some point you need to present a rigorous error analysis, i.e. to estimate the retrieval uncertainty separately for each error source. For this reason, I would suggest switching to "optimal estimation" (a.k.a. 1DVAR) following Rodgers. I don't see any fundamental reason why that should not be possible.

Kind regards,

Patrick Eriksson